# Efficient Planning with Latent Diffusion

**Wenhao Li**

School of Software Engineering, Tongji University

Shanghai, 201804, China

`liwenhao@cuhk.edu.cn`

## Abstract

Temporal abstraction and efficient planning pose significant challenges in offline reinforcement learning, mainly when dealing with domains that involve temporally extended tasks and delayed sparse rewards. Existing methods typically plan in the raw action space and can be inefficient and inflexible. Latent action spaces offer a more flexible paradigm, capturing only possible actions within the behavior policy support and decoupling the temporal structure between planning and modeling. However, current latent-action-based methods are limited to discrete spaces and require expensive planning steps. This paper presents a unified framework for continuous latent action space representation learning and planning by leveraging latent, score-based diffusion models. We establish the theoretical equivalence between planning in the latent action space and energy-guided sampling with a pretrained diffusion model and incorporate a novel sequence-level exact sampling method. Our proposed method, `LatentDiffuser`, demonstrates competitive performance on low-dimensional locomotion control tasks and surpasses existing methods in higher-dimensional tasks.

## 1 Introduction

A considerable volume of samples gathered by operational systems gives rise to the issue of offline reinforcement learning (RL), specifically, the recovery of high-performing policies without additional environmental exploration (Wu et al., 2019; Kumar et al., 2020; Kostrikov et al., 2021; 2022; Ghosh et al., 2022). However, domains that encompass temporally extended tasks and severely delayed sparse rewards can present a formidable challenge for standard offline approaches (Li et al., 2015; Ren et al., 2021; Li et al., 2023). Analogous to the online setting, an emergent objective in offline RL involves the development of efficacious hierarchy methodologies that can obtain temporally extended lower-level primitives, subsequently facilitating the construction of a higher-level policy operating at a more abstract temporal scale (Ajay et al., 2021; Pertsch et al., 2021; Villecroze et al., 2022; Rosete-Beas et al., 2022; Rao et al., 2022; Yang et al., 2023).

Within the hierarchical framework, current offline RL approaches can be broadly categorized into *model-free* and *model-based*. The former conceptualizes the higher-level policy optimization as a auxilary offline RL issue (Liu et al., 2020; Liu & Sun, 2022; Ma et al., 2022; Kipf et al., 2019; Ajay et al., 2021; Rosete-Beas et al., 2022). In contrast, the latter encompasses planning in the higher-level policy space by generating future trajectories through a dynamics model of the environment, either predefined or learned (Li et al., 2022; Co-Reyes et al., 2018; Lynch et al., 2020; Lee et al., 2022; Venkatraman, 2023). Concerning lower-level primitive learning, these two methods exhibit similarities and are typically modeled as goal-conditioned or skill-based imitation learning or offline RL problems. Conversely, the instabilities arising from offline hierarchical RL methodologies due to the "deadly triad (Sutton & Barto, 2018; Van Hasselt et al., 2018)," restricted data access (Fujimoto et al., 2019; Kumar et al., 2020), and sparse rewards (Andrychowicz et al., 2017; Ma et al., 2022) remain unaddressed. This spawns another subset of model-based approaches along with more effective hierarchical variants that endeavor to resolve problems from a sequence modeling viewpoint Chen et al. (2021); Janner et al. (2021; 2022); Ajay et al. (2023).

Irrespective of whether a method is model-free or model-based, it adheres to the traditional settings, wherein planning occurs in the raw action space of the Markov Decision Process (MDP). Although

seemingly intuitive, planning in raw action space can be inefficient and inflexible (Wang et al., 2020; Yang et al., 2021; Jiang et al., 2023). Challenges include ensuring model accuracy across the entire space and the constraint of being tied to the environment's temporal structure. Conversely, human planning offers enhanced flexibility through temporal abstractions, high-level actions, backward planning, and incremental refinement.

Drawing motivation from TAP (Jiang et al., 2023), we put forth the notion of the *latent action*. Planning within the domain of latent actions delivers a twofold advantage compared to planning with raw actions. Primarily, it encompasses only plausible actions under behavior policy support, yielding a reduced space despite the raw action space's dimensionality and preventing the exploitation of model frailties. Secondarily, it permits the separation of the temporal structure between planning and modeling, thus enabling a more adaptable and efficient planning process unconstrained by specific transitions. These dual benefits render latent-action-based approaches naturally superior to extant methodologies when handling temporally extended offline tasks.

Nevertheless, two shortcomings of TAP inhibit its ability to serve as a general and practical framework. Initially, TAP is confined to *discrete* latent action spaces. In real-world contexts, agents are likely to carry out a narrow, discrete assortment of tasks and a broader spectrum of behaviors (Co-Reyes et al., 2018). This introduces a predicament — should a minor skill modification be necessary, such as opening a drawer by seizing the handle from top to bottom instead of bottom to top, a completely novel set of demonstrations or reward functions might be mandated for behavior acquisition. Subsequently, once the latent action space has been ascertained, TAP necessitates a distinct, resource-intensive planning phase for generating reward-maximizing policies. The price of planning consequently restricts latent actions to discrete domains.

To tackle these limitations, this paper proposes a novel framework, `LatentDiffuser`, by concurrently modeling *continuous* latent action space representation learning and latent action-based planning as a conditional generative problem within the latent domain. Specifically, `LatentDiffuser` employs unsupervised techniques to discern the latent action space by utilizing score-based diffusion models (SDMs) (Song et al., 2021; Nichol & Dhariwal, 2021; Ho & Salimans, 2022) within the latent sphere in conjunction with a variational autoencoder (VAE) framework (Kingma & Welling, 2014; Rezende et al., 2014; Vahdat et al., 2021). We first segment the input trajectories, map each slice to latent action space (which needs to be learned), and apply the SDM to the latent sequence. Subsequently, the SDM is entrusted with approximating the distribution over the offline trajectory embeddings, conditioned on the related return values. Planning—or reward-maximizing trajectory synthesis—is realized by initially producing latent actions through sampling from a simple base distribution, followed by iterative, conditional denoising, and eventually translating latent actions into the trajectory space using a decoder. In other words, `LatentDiffuser` can be regarded as a VAE equipped with an SDM prior (Vahdat et al., 2021).

Theoretically, we demonstrate that planning in the domain of latent actions is tantamount to energy-guided sampling using a pre-trained diffusion behavior model. Exact energy-guided sampling is essential to carry out high-quality and efficient planning. To achieve this objective, we modify QGPO (Lu et al., 2023) to realize exact sampling at the sequence level. Comprehensive numerical results on low-dimensional locomotion control tasks reveal that `LatentDiffuser` exhibits competitive performance against robust baselines and outperforms them on tasks of greater dimensionality. Our main contributions encompass: 1) Developing a unified framework for continuous latent action space representation learning and planning that delivers flexibility and efficiency in temporally extended offline decision-making. 2) Our theoretical derivation confirms the equivalence between planning in the latent action space and energy-guided sampling with a pretrained diffusion model. It introduces an innovative sequence-level exact sampling technique. 3) Numerical experiments exhibit the competitive performance of `LatentDiffuser` and its applicability across a range of low- and high-dimensional continuous control tasks.

## 2 RELATED WORK

Owing to spatial constraints, this section will briefly present the most pertinent domain of `LatentDiffuser`: offline RL or imitation learning (IL) based on a hierarchical structure. In terms of algorithmic specificity, existing techniques can be broadly classified into *goal-based* and *skill-based* methods (Pateria et al., 2021). For further related literature, including but not limited to

model-based RL, action representation learning, offline RL, and RL as sequence modeling, kindly refer to Appendix C and the appropriate citations within the papers.

*Goal-based* approaches primarily concentrate on attaining a designated state. The vital aspect of such techniques concerns the selection or creation of subgoals, which reside in the raw state space. Once the higher-level subgoal is ascertained, the lower-level policy is generally acquired through standard IL methods or offline RL based on subgoal-augmented/conditioned policy, a universal value function (Schaul et al., 2015), or their combination. In extant methods, the subgoal is either predefined (Zhou et al., 2019; Xie et al., 2021; Ma et al., 2021), chosen based on heuristics (Ding et al., 2014; Guo & Zhai, 2016; Pateria et al., 2020; Mandlekar et al., 2020), or generated via planning or an additional offline RL technique (Liu et al., 2020; Liu & Sun, 2022; Li et al., 2022; Ma et al., 2022). Moreover, some methods (Eysenbach et al., 2019; Paul et al., 2019; Lai et al., 2020; Kujanpää et al., 2023) are solely offline during the subgoal selection or generation process. This paper also pertains to the options framework (Sutton et al., 1999; Stolle & Precup, 2002; Bacon et al., 2017; Wulfmeier et al., 2021; Salter et al., 2022; Villecroze et al., 2022)), as both the (continuous) latent actions of `LatentDiffuser` and (discrete) options introduce a mechanism for temporal abstraction.

*Skill-based* methods embody higher-level skills as low-dimensional latent codes. In this context, a skill signifies a subtask's policy, semantically representing "the capability to perform something adeptly" (Pateria et al., 2021). Analogous to goal-based approaches, once the higher-level skill is identified, the lower-level skill-conditioned policy is generally acquired through standard IL or offline RL methods. More precisely, few works utilize predefined skills (Nasiriany et al., 2022; Fatemi et al., 2022). The majority of studies employ a two- or multi-phase training framework: initially, state sequences are projected into continuous latent variables (i.e., skills) via unsupervised learning; next, optimal skills are generated based on offline RL (Kipf et al., 2019; Pertsch et al., 2021; Ajay et al., 2021; Rosete-Beas et al., 2022; Lee et al., 2022; Venkatraman, 2023) or planning[1] (Co-Reyes et al., 2018; Lynch et al., 2020; Lee et al., 2022; Venkatraman, 2023) in the skill space.

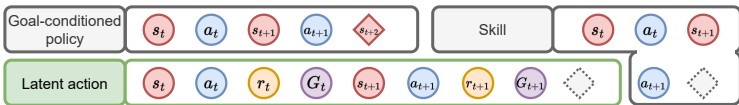

Figure 1: The physical meaning of the goal-conditioned policy, skill and latent action (corresponding to 2 timesteps in the raw MDP). The red diamond represents a particular (goal) state, the gray, dotted diamond is a placeholder, and the red circle denotes any state.

In contrast with the aforementioned hierarchical methodologies, `LatentDiffuser` initially learns a more compact latent action space and subsequently employs the latent actions to make decisions. As demonstrated in Figure 1, latent action not only differs from the goal-conditioned policy, which pertains to the trajectory of reaching a particular state, but also from the skill, which relates to the trajectory of completing a specific (multi-step) state transition. The latent action also corresponds to the agent's received reward and the subsequent expected return. The unique physical implications of latent action and the methodology utilized by `LatentDiffuser` render the proposed method advantageous in several ways. 1) The future information in the latent action allows the algorithm to execute more efficient planning. 2) Unlike existing works wherein multiple optimization objectives and the fully coupling or seperating of representation learning and decision making (RL or planning) lead to intricate training processes and reduced training efficiency, `LatentDiffuser` exhibits end-to-end training and unifies representation learning, sampling, and planning.

## 3 PROBLEM FORMULATION

In this paper, we approach the offline RL problem as a sequence modeling task, in alignment with previous work (Janner et al., 2022; Ajay et al., 2023; Li et al., 2023). The following subsection delineates the specificities of sequence modeling, or more accurately, the conditional generative modeling paradigm. We examine a trajectory, $\tau$, of length $T$, which is sampled from a MDP that features a

---

[1]It is important to note that planning is only feasible when the environment model is known or can be sampled from the environment model. Consequently, some of these works focus on online RL tasks, while others first learn an additional environment model from the offline dataset and then plan in the skill space.

fixed stochastic behavior policy. This trajectory comprises (refer to Appendix for more modeling selections of $\tau$) a series of states, actions, rewards, and reward-to-go values, $G_t := \sum_{i=t} \gamma^{i-t} r_i$, as proxies for future cumulative rewards: $\tau := (s_1, a_1, r_1, G_1, s_2, a_2, r_2, G_2, \ldots, s_T, a_T, r_T, G_T)$. It is crucial to note that the definition of $\tau$ diverges from that in prior studies (Janner et al., 2022; Ajay et al., 2023; Li et al., 2023), as each timestep now contains both the reward and reward-to-go values. This modification has been specifically engineered to facilitate the subsequent learning of latent action spaces. Sequential decision-making is subsequently formulated as the standard problem of conditional generative modeling:

$$\max_{\theta} \mathbb{E}_{\tau \sim \mathcal{D}} \left[ \log p_\theta \left( \tau_0 \mid \boldsymbol{y}(\tau_0) \right) \right], \tag{1}$$

where $\tau_0 := \tau$. The objective is to estimate the conditional trajectory distribution using $p_\theta$ to enable planning or generating the desired trajectory $\tau_0$ based on the information $\boldsymbol{y}(\tau_k)$. Existing instances of $\boldsymbol{y}$ may encompass the return (Janner et al., 2022; Li et al., 2023), the constraints met by the trajectory (Ajay et al., 2023; Li et al., 2023), or the skill demonstrated in the trajectory (Ajay et al., 2023). The generative model is constructed in accordance with the conditional diffusion process:

$$q \left( \tau_{k+1} \mid \tau_k \right), \quad p_\theta \left( \tau_{k-1} \mid \tau_k, \boldsymbol{y}(\tau_0) \right). \tag{2}$$

As per standard convention, $q$ signifies the forward noising process while $p_\theta$ represents the reverse denoising process (Ajay et al., 2023).

**Latent Actions**  We introduce the concept of *latent action* (Figure 1) proposed in TAP (Jiang et al., 2023). TAP specifically models the *optimal* conditional trajectory distribution $p^*(\tau \mid s_1, \boldsymbol{z})$ using a series of latent variables, $\boldsymbol{z} := (z_1, \ldots, z_M)$. Assuming that the state and latent variables $(s_1, \boldsymbol{z})$ can be deterministically mapped to trajectory $\tau$, $p^*(\tau \mid s_1, \boldsymbol{z}) := p(s_1)\mathbb{1}(\tau = h(s_1, \boldsymbol{z}))\pi^*(\boldsymbol{z} \mid s_1)$ is obtained. The terms $\boldsymbol{z}$ and $\pi^*(\boldsymbol{z} \mid s_1)$ are subsequently referred to as the *latent actions* and the *optimal latent policy*, respectively. In a *deterministic* MDP, the trajectory corresponding to an arbitrary function $h(s_1, \boldsymbol{z})$ with $\pi^*(\boldsymbol{z} \mid s_1) > 0$ will constitute an optimal executable plan, implying that the optimal trajectory can be recovered by following the latent actions $\boldsymbol{z}$, beginning from the initial state $s_1$. Consequently, planning within the latent action space $\mathcal{Z}$ facilitates the discovery of an desired, optimal trajectory. TAP, however, remain restricted to discrete latent action spaces and necessitate independent, resource-intensive planning. Motivated by these limitations, we present a unified framework that integrates representation learning and planning for continuous latent action via latent, score-based diffusion models.

# 4 ALGORITHM FRAMEWORK

This section provides a comprehensive elaboration of the model components and design choices, such as the network architecture, loss functions, as well as the details of training and planning. By unifying the representation learning and planning of latent action through the incorporation of a latent diffusion model and the exact energy-guided sampling technique, `LatentDiffuser` achieves effective decision-making capabilities for temporally-extended, sparse reward tasks. Specifically, we first explore the representation learning for latent action in Section 4, followed by a detailed discussion on planning using energy-guided sampling in Section 4.2, and provide a algorithm summary in Section 4.3 to close this section.

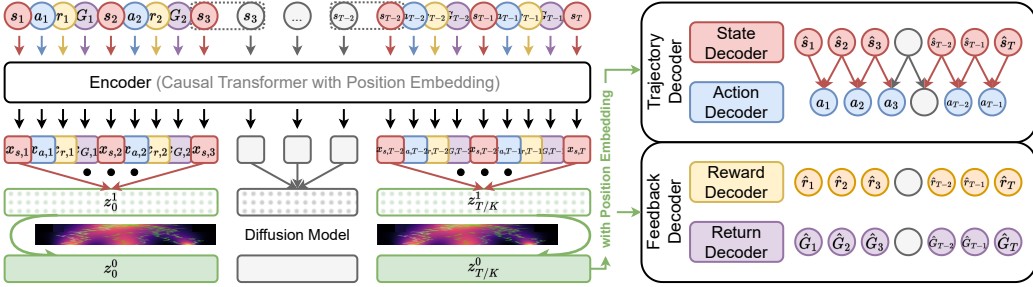

Figure 2: Representation learning for latent action with the latent score-based diffusion model.

## 4.1 REPRESENTATION LEARNING FOR LATENT ACTION

The latent action space allows for a more compact, efficient, and adaptable method by effectively capturing behavior policy support and detaching the temporal structure, thus providing innate benefits in handling temporally extended offline tasks. As indicated in Section 3, before proceeding to planning, we must first learn a continuous latent action space. For this purpose, we propose the `LatentDiffuser` based on a latent diffusion model (LDM) (Vahdat et al., 2021), as depicted in Figure 2. `LatentDiffuser` is constituted by an encoder $q_\phi (z_0 \mid s_1, \tau)$, a score-based prior $p_\theta (z_0 \mid s_1)$, and a decoder $p_\psi (\tau \mid s_1, z_0)$. In accordance with Vahdat et al. (2021), we train `LatentDiffuser` by minimizing the variational upper bound on the negative trajectory log-likelihood $\log p(\tau \mid s_1)$, meaning that the information $y(\tau)$ in Equation (1) is instantiated as the initial state $s_1$:

$$\begin{aligned}
\mathcal{L}(s_1, \tau, \phi, \theta, \psi) &= \mathbb{E}_{q_\phi(\mathbf{z}_0 \mid s_1, \tau)} \left[ -\log p_\psi (\tau \mid s_1, \mathbf{z}_0) \right] + \mathrm{KL} \left( q_\phi (\mathbf{z}_0 \mid s_1, \tau) \| p_\theta (\mathbf{z}_0 \mid s_1) \right) \\
&= \mathbb{E}_{q_\phi(\mathbf{z}_0 \mid s_1, \tau)} \left[ -\log p_\psi (\tau \mid s_1, \mathbf{z}_0) \right] + \mathbb{E}_{q_\phi(\mathbf{z}_0 \mid s_1, \tau)} \left[ \log q_\phi (\mathbf{z}_0 \mid s_1, \tau) \right] \quad (3) \\
&\quad + \mathbb{E}_{q_\phi(\mathbf{z}_0 \mid s_1, \tau)} \left[ -\log p_\theta (\mathbf{z}_0 \mid s_1) \right]
\end{aligned}$$

utilizing a VAE approach (Kingma & Welling, 2014; Rezende et al., 2014), wherein the $q_\phi (\mathbf{z}_0 \mid s_0, \tau)$ approximates the true posterior $p(\mathbf{z}_0 \mid s_0, \tau)$.

This paper employs Equation (3) with a decomposed KL divergence into entropy and cross entropy terms. The reconstruction and entropy terms are easily estimated for any explicit encoder as long as the reparameterization trick is applicable (Kingma & Welling, 2014). The challenging aspect of training `LatentDiffuser` pertains to training the cross entropy term, which involves the score-based prior. Unlike Vahdat et al. (2021), which addresses this challenge by simultaneously learning an encoder/decoder architecture alongside a score-based prior, we adopt a simpler yet efficacious approach (Rombach et al., 2022) by training a VAE $\{q_\phi, p_\psi\}$ and a score-based diffusion model $\{q_\theta\}$ consecutively based on the offline dataset $\mathcal{D}_\tau$. This does not necessitate a delicate balancing of reconstruction and generative capabilities.

**Encoder $q_\phi$ and Decoder $p_\psi$**   We use the almost consistent encoder design with TAP (Jiang et al., 2023). Specifically, we handle $x_t := (s_t, a_t, r_t, G_t)$ as a single token. The encoder $\phi$ processes token $x_t$ using a GPT-2 style Transformer[2], yielding $T$ feature vectors, where $T$ is the episode horizon. Subsequently, we apply a 1-dimensional max pooling with a kernel size and stride of $L$, followed by a linear layer, and generate $T/L$ latent actions. Moreover, different from the TAP Decoder architecture, we use a modular design idea. More concretely, each latent action is tiled $L$ times to match the number of input/output tokens $T$. We then concatenate the initial state $s_1$ and the latent action, and apply a linear projection to provide state information to the decoder. After adding positional embedding, the decoder reconstructs the trajectory $\hat{\tau} := (\hat{x}_1, \hat{x}_2, \ldots, \hat{x}_T)$, with $\hat{x}_t := (\hat{s}_t, \hat{a}_t, \hat{r}_t, \hat{G}_t)$. To enhance the decoder's representation ability, we design the decoder modularly for different elements in $x_t$, as shown in Figure 2. Noting that the action decoder is designed based on the inverse dynamics model (Agrawal et al., 2015; Pathak et al., 2017) in a manner similar to (Ajay et al., 2023; Li et al., 2023), with the aim of generating raw action sequences founded on the state sequences. The training of the encoder and decoders finally entails the use of a reconstruction loss computed as the mean squared error between input trajectories $\{\tau\}$ and reconstructed trajectories $\{\hat{\tau}\}$, coupled with a low-weighted ($\approx 10^{-6}$) Kullback-Leibler penalty towards a standard normal on the learned latent actions, akin to VAE approaches (Kingma & Welling, 2014; Rezende et al., 2014). This prevents the arbitrary scaling of latent action space.

**Score-based Prior $\theta$**   Having trained the VAE $\{q_\phi, p_\psi\}$, we now have access to a compact latent action space. Distinct from VAE's adoption of a uniform prior or TAP's utilization of an autoregressive, parameterized prior over latent actions, `LatentDiffuser` employs a score-based one. Thus, by harnessing the "diffusion-sampling-as-planning" framework, we seamlessly transform planning into conditional diffusion sampling, ultimately circumventing the need for an independent, costly planning stage. Concretely, the score-based prior is modeled as a conditional, score-based diffusion probabilistic model, which is parameterized using a temporal U-Net architecture (Janner et al., 2022; Ajay et al., 2023). This architecture effectively treats a sequence of noised latent action $x_k(z)$ as an image, where the height represents a single latent action's dimension and the width signifies

---

[2]Different from the casual Transformer used in TAP, see Appendix for more discussion.

the number of the latent actions. Conditioning information $\boldsymbol{y}(\boldsymbol{z}) := s_1$ is then projected using a multi-layer perceptron (MLP). The training of the score-based prior is formulated as a standard score-matching problem detailed in Appendix B.2.

## 4.2 PLANNING WITH ENERGY-GUIDED SAMPLING

Upon acquiring the latent action space, we are able to effectively address temporally-extended offline tasks using planning. Intriguingly, when examined from a probabilistic standpoint, the optimal latent action sequence sampling coincides with a guided diffusion sampling problem (Lu et al., 2023), wherein the guidance is shaped by an (unnormalized) energy function. By adopting a "diffusion-sampling-as-planning" framework (Janner et al., 2022), we can perform planning through conditional sampling using the pretrained LatentDiffuser, without necessitating further costly planning steps (Janner et al., 2021; Jiang et al., 2023). This renders LatentDiffuser a holistic framework that seamlessly consolidates representation learning and planning within the latent action space. In the subsequent sections, the equivalence between optimal latent actions sampling and energy-guided diffusion sampling is demonstrated, followed by the introduction of a practical sampling algorithm to facilitate efficient planning.

**Planning is Energy-Guided Diffusion Sampling**    Considering a deterministic mapping from $\tau$ to $\boldsymbol{z}$, achieved by the learned encoder $q_\phi$, the following theorem (refer to Appendix I.1 for the proof) is derived for the *optimal latent policy* defined in Section 3:

> **Theorem 1** (Optimal latent policy). *Given an initial state $s_1$, the optimal latent policy satisfies: $\pi^*(\boldsymbol{z} \mid s_1) \propto \mu(\boldsymbol{z} \mid s_1)e^{\beta \sum_{t=1}^{T} Q_\zeta(s_t, a_t)}$, wherein $\mu(\boldsymbol{z} \mid s_1)$ represents the behavior latent policy and $Q_\zeta(\cdot, \cdot)$ refers to the estimated Q-value function. $\beta \geq 0$ signifies the inverse temperature controlling the energy strength.*

By rewriting $p_0 := \pi^*$, $q_0 = \mu$ and $\boldsymbol{z}_0 = \boldsymbol{z}$, we can reformulate the optimal planning into the following diffusion sampling problem:

$$p_0(\boldsymbol{z}_0 \mid s_1) \propto q_0(\boldsymbol{z}_0 \mid s_1) \exp\left(-\beta \mathcal{E}(h(\boldsymbol{z}_0, s_1))\right), \tag{4}$$

where $\mathcal{E}(h(\boldsymbol{z}_0, s_1)) := -\sum_{t=1}^{T} Q_\zeta(s_t, a_t)$ and $h(\boldsymbol{z}_0, s_1)$ denotes the pretrained decoder $p_\psi$. The behavior latent policy $q_0(\boldsymbol{z}_0 \mid s_1)$ is modeled by the pretrained LatentDiffuser. We then adopt the "diffusion-sampling-as-planning" to generate desired (e.g., reward-maximizing) latent actions $\boldsymbol{z}_0$. Concretely, we employ $q_0 := q$, $p_0 = p$ at diffusion timestep $k = 0$. Then a forward diffusion process is constructed to simultaneously diffuse $q_0$ and $p_0$ into an identical noise distribution, where $p_{k0}(\boldsymbol{z}_k|\boldsymbol{z}_0, s_1) := q_{k0}(\boldsymbol{z}_k|\boldsymbol{z}_0, s_1) = \mathcal{N}(\boldsymbol{z}_k|\alpha_k \boldsymbol{z}_0, \sigma_t^2 \mathbf{I})$. Based on (Lu et al., 2023, Theorem 3.1), the marginal distribution $q_k$ and $p_k$ of the noised latent actions $\boldsymbol{z}_k$ at the diffusion timestep $k$ adhere to:

$$p_k(\boldsymbol{z}_k \mid s_1) \propto q_k(\boldsymbol{z}_k \mid s_1) \exp\left(\mathcal{E}_k(h(\boldsymbol{z}_k, s_1))\right), \tag{5}$$

where $\mathcal{E}_k(h(\boldsymbol{z}_k, s_1))$ is $\beta \mathcal{E}(h(\boldsymbol{z}_0, s_1))$ when $k = 0$ and $-\log \mathbb{E}_{q_{0k}(\boldsymbol{z}_0|\boldsymbol{z}_k)}[\exp(-\beta \mathcal{E}(h(\boldsymbol{z}_0, s_1)))]$ when $k > 0$. We then need to estimate the score function of $p_k(\boldsymbol{z}_k \mid s_1)$. Quoting the derivation of Lu et al. (2023), the score function satisfies: $\nabla_{\boldsymbol{z}_k} \log p_k(\boldsymbol{z}_k \mid s_1) = \nabla_{\boldsymbol{z}_k} \log q_k(\boldsymbol{z}_k \mid s_1) + \nabla_{\boldsymbol{z}_k} \mathcal{E}_k(h(\boldsymbol{z}_k, s_1))$. Consequently, the optimal planning has been formulated as energy-guided sampling within the latent action space, with $\nabla_{\boldsymbol{z}_k} \mathcal{E}(h(\boldsymbol{z}_k, s_1))$ as the desired guidance.

**Practical Sampling Method**    Estimating the target score function $\nabla_{\boldsymbol{z}_k} \log p_k(\boldsymbol{z}_k \mid s_1)$ is non-trivial because of the intractable energy guidance $\nabla_{\boldsymbol{z}_k} \mathcal{E}(h(\boldsymbol{z}_k, s_1))$. We borrow the energy-guided sampling method proposd in (Lu et al., 2023) and propose a sequence-level, exact sampling methods by training a total of three neural networks: (1) a diffusion model to model the behavior latent policy $q_0(\boldsymbol{z}_0 \mid s_1)$; (2) a state-action value function $Q_\zeta(s, a)$ to define the intermediate energy function $\mathcal{E}(h(\boldsymbol{z}_0, s_1))$; and (3) an time-dependent energy model $f_\eta(\boldsymbol{z}_k, s_1, k)$ to estimate $\mathcal{E}_k(h(\boldsymbol{z}_k, s_1))$ and guide the diffusion sampling process.

Recall that we already have (1) a diffusion model, i.e., the socre-based prior $p_\theta(\boldsymbol{z}_0 \mid s_1)$ and (2) a state-action value function $Q_\zeta(s, a)$, i.e., the return decoder. According to Lu et al. (2023, Theorem 3.2), the only remained time-dependent energy model, $f_\eta(\boldsymbol{z}_k, s_1, k)$, can be trained by minizing the

following contrastive loss:

$$\min_{\eta} \mathbb{E}_{p(k,s1)} \mathbb{E}_{\prod_{i=1}^{M} q\left(\boldsymbol{z}_0^{(i)} | s_1\right) p(\epsilon^{(i)})} \left[ -\sum_{i=1}^{M} \frac{e^{-\beta \mathcal{E}_0(h(\boldsymbol{z}_k^{(i)}, s_1))}}{\sum_{j=1}^{M} e^{-\beta \mathcal{E}_0(h(\boldsymbol{z}_0^{(j)}, s_1))}} \log \frac{e^{f_{\eta}\left(\boldsymbol{z}_k^{(i)}, s_1, k\right)}}{\sum_{j=1}^{M} e^{f_{\eta}\left(\boldsymbol{z}_k^{(j)}, s_1, k\right)}} \right],$$
(6)

where $k \sim \mathcal{U}(0, K)$, $\boldsymbol{z}_k = \alpha_k \boldsymbol{z}_0 + \sigma_k \epsilon$, and $\epsilon \sim \mathcal{N}(0, \mathbb{I})$. To estimate true latent actions distribution $q(\boldsymbol{z}_0 \mid s_1)$ in Equation 6, we utilize the pretrained encoder $q_{\phi}$ and score-based prior $p_{\theta}$ to generate $M$ support latent actions $\{\hat{\boldsymbol{z}}_0^{(i)}\}_M$ for each initial state $s_1$ by diffusion sampling. The contrastive loss in Equation (6) is then estimated by:

$$\min_{\eta} \mathbb{E}_{k,s_1,\epsilon} - \sum_{i=1}^{M} \frac{e^{-\beta \mathcal{E}_0(h(\hat{\boldsymbol{z}}_0^{(i)}, s_1))}}{\sum_{j=1}^{M} e^{-\beta \mathcal{E}_0(h(\hat{\boldsymbol{z}}_0^{(j)}, s_1))}} \log \frac{e^{f_{\eta}\left(\hat{\boldsymbol{z}}_k^{(i)}, s_1, k\right)}}{\sum_{j=1}^{M} e^{f_{\eta}\left(\hat{\boldsymbol{z}}_k^{(j)}, s_1, k\right)}},$$
(7)

where $\hat{\boldsymbol{z}}_0^{(i)}, \hat{\boldsymbol{z}}_0^{(j)}$ correspond to the support latent actions for each initial state $s_1$.

### 4.3 ALGORITHM SUMMARY

In general, the training phase of `LatentDiffuse` is composed of three parts, corresponding to the training of encoder and decoders $\{q_{\phi}, p_{\psi}\}$, score-based prior $p_{\theta}$, and intermediated energy model $f_{\eta}$, as shown in Algorithm 1. Throughout the training process, it is imperative to employ two distinct datasets: the first being a standard offline RL dataset, $\mathcal{D}$, which encompasses trajectories sampled from behavior policies, whereas the second dataset consists of support latent actions for each initial state $s_1 \in \mathcal{D}$, generated by the pre-trained VAE, i.e., the encoder, score-based prior and decoders.

---

**Algorithm 1** `LatentDiffuser`: Efficient Planning with Latent Diffusion

---

Initialize the latent diffusion model, i.e., the encoder $q_{\phi}$, the score-based prior $p_{\theta}$ and the decoder $p_{\psi}$; the intermediate energy model $f_{\eta}$
**for** each gradient step **do**                                                 ▷ Training the encoder and decoders
    Sample $B_1$ trajectories $\tau$ from offline dataset $\mathcal{D}$
    Generate reconstructed trajectories $\hat{\tau}$ with the encoder $q_{\phi}$ and decoder $p_{\psi}$
    Update $\{\phi, \psi\}$ based on the standard *VAE loss*
**end for**
**for** each gradient step **do**                                                       ▷ Training the score-based prior
    Sample $B_2$ trajectories $\tau$ from offline dataset $\mathcal{D}$
    Sample $B_2$ Gaussian noises $\epsilon$ from $\mathcal{N}(0, \mathbf{I})$ and $B_2$ time $k$ from $\mathcal{U}(0, K)$
    Generate latent actions $\boldsymbol{z}_0$ with the pretrained encoder $q_{\phi}$ and decoder $p_{\psi}$
    Perturb $\boldsymbol{z}_0$ according to $\boldsymbol{z}_k := \alpha_k \boldsymbol{z}_0 + \sigma_k \epsilon$
    Update $\{\theta\}$ with the standard *score-matching loss* in Appendix B.2
**end for**
**for** each initial state $s_1$ in offline dataset $\mathcal{D}$ **do**          ▷ Generating the support latent actions
    Sample $M$ support latent actions $\{\hat{\boldsymbol{z}}_0^{(i)}\}_M$ from the pretrained score-based prior $p_{\theta}$
**end for**
**for** each gradient step **do**                                               ▷ Training the intermediate energy model
    Sample $B_3$ initial state $s_1$ from offline dataset $\mathcal{D}$
    Sample $B_3$ Gaussian noises $\epsilon$ from $\mathcal{N}(0, \mathbf{I})$ and $B_3$ time $k$ from $\mathcal{U}(0, K)$
    Retrieve support latent actions $\{\hat{\boldsymbol{z}}_0^{(i)}\}_M$ for each $s_1$
    Perturb $\hat{\boldsymbol{z}}_0^{(i)}$ according to $\hat{\boldsymbol{z}}_k^{(i)} := \alpha_k \hat{\boldsymbol{z}}_0^{(i)} + \sigma_k \epsilon$
    Update $\{\eta\}$ based on the *contrastive loss* in Equation (7)
**end for**

---

Moreover, the optimal planning is tantamount to conducting conditional diffusion sampling based on the score-based prior and the intermediate energy model. Formally, the generation employs reverse denoising process at each diffusion timestep $k$ by utilizing the score function $\nabla_{\boldsymbol{z}_k} \log p_k (\boldsymbol{z}_k \mid s_1)$ based on the score function of the score-based prior $\nabla_{\boldsymbol{z}_k} \log q_k (\boldsymbol{z}_k \mid s_1)$ and intermediate energy model $\nabla_{\boldsymbol{z}_k} \mathcal{E}_k (h(\boldsymbol{z}_k, s_1))$, along with the state and action decoder $p_{\psi} (\tau \mid s_1, \boldsymbol{z}_0)$ to map the sampled latent actions $\boldsymbol{z}_0$ back to the original trajectory space. Explicitly, the generative process is

$p(s_1, \boldsymbol{z}_0, \tau) = p_0(\boldsymbol{z}_0 \mid s_1) p_\psi(\tau \mid s_1, \boldsymbol{z}_0)$. To avoid the accumulation of errors during sampling, we adopt the *receding horizon control* used in the existing methods (Ajay et al., 2023; Li et al., 2023).

## 5 EXPERIMENTS

This section aims to assess the efficacy of the `LatentDiffuser` for extended temporal offline tasks in comparison to current SOTA offline RL methods, which integrate hierarchical structures, and conditional generation models. The empirical evaluation encompasses three task categories derived from D4RL (Fu et al., 2020): namely, *Gym locomotion control*, *Adroit*, and *AntMaze*. Gym locomotion tasks function as a proof-of-concept in the lower-dimensional realm, in order to ascertain whether `LatentDiffuser` is capable of accurately reconstructing trajectories for decision-making and control purposes. Subsequently, `LatentDiffuser` is evaluated on Adroit—a task with significant state and action dimensionality—as well as on `LatentDiffuser` within the AntMaze environment, which represents a sparse-reward continuous-control challenge in a series of extensive long-horizon maps (Li et al., 2023). The subsequent sections will describe and examine the performance of these tasks and their respective baselines individually. Scores within 5% of the maximum per task will be emphasized in bold (Kostrikov et al., 2022).

### 5.1 PROOF-OF-CONCEPT: GYM LOCOMOTION CONTROL

**Baselines** Initially, an outline of the baselines is provided: CQL (Kumar et al., 2020), IQL (Kostrikov et al., 2022), D-QL (Wang et al., 2023), and QGPO (Lu et al., 2023) are all model-free offline RL methods. MoReL (Kidambi et al., 2020) is a model-based offline RL method. DT (Chen et al., 2021), TT (Janner et al., 2021), Diffuser (Janner et al., 2022), and DD (Ajay et al., 2023) address offline RL tasks via conditional generative modeling. Finally, TAP (Jiang et al., 2023) and HDMI (Li et al., 2023) employ a hierarchical framework grounded in generative modeling. Due to spatial constraints, only algorithms with the highest performance rankings are displayed herein; for a comprehensive comparison, please refer to the appendix.

Table 1: The performance in Gym locomotion control in terms of normalized average returns. Results correspond to the mean and standard error over 5 planning seeds.

| Dataset | Environment | CQL | TT | DD | D-QL | TAP | QGPO | HDMI | LD |
|---------|-------------|-----|----|----|----|-----|------|------|-----|
| Med-Expert | HalfCheetah | 91.6 | **95** | 90.6±1.3 | **96.8±0.3** | 91.8 ± 0.8 | **93.5±0.3** | 92.1±1.4 | **95.2±0.2** |
| Med-Expert | Hopper | 105.4 | 110.0 | 111.8±1.8 | 111.1±1.3 | 105.5 ± 1.7 | 108.0±2.5 | **113.5±0.9** | 112.9±0.3 |
| Med-Expert | Walker2d | **108.8** | 101.9 | **108.8±1.7** | 110.1±0.3 | 107.4 ± 0.9 | 110.7 ± 0.6 | 107.9±1.2 | 111.3±0.2 |
| Medium | HalfCheetah | 44.0 | 46.9 | 49.1±1.0 | 51.1±0.5 | 45.0 ± 0.1 | **54.1 ± 0.4** | 48.0±0.9 | **53.6±0.4** |
| Medium | Hopper | 58.5 | 61.1 | 79.3±3.6 | 90.5±4.6 | 63.4 ± 1.4 | **98.0 ± 2.6** | 76.4±2.6 | **98.5±0.7** |
| Medium | Walker2d | 72.5 | 79 | 82.5±1.4 | **87.0±0.9** | 64.9 ± 2.1 | **86.0 ± 0.7** | 79.9±1.8 | **86.3±0.9** |
| Med-Replay | HalfCheetah | **45.5** | 41.9 | 39.3±4.1 | **47.8±0.3** | 40.8 ± 0.6 | **47.6 ± 1.4** | 44.9±2.0 | **47.3±1.2** |
| Med-Replay | Hopper | 95 | 91.5 | **100±0.7** | 101.3±0.6 | 87.3 ± 2.3 | **96.9 ± 2.6** | 99.6±1.5 | **100.4±0.5** |
| Med-Replay | Walker2d | 77.2 | **82.6** | 75±4.3 | **95.5±1.5** | 66.8 ± 3.1 | 84.4 ± 4.1 | 80.7±2.1 | 82.6 ± 2.1 |
| **Average** | | 77.6 | 78.9 | 81.8 | **88.0** | 82.5 | **86.6** | 82.6 | **87.5** |

Table 1 shows that `LatentDiffuser` surpasses specifically designed offline RL methods in the majority of tasks. Furthermore, the performance discrepancy between `LatentDiffuser` and two-stage algorithms, such as TAP and HDMI, underscores the benefits provided by the the proposed framework, which unifies learning of latent action space representation and planning.

### 5.2 HIGH-DIMENSIONAL MDP: ADROIT

**Baselines** Taking into account the large dimensions characterizing the Adroit task actions, only baselines that perform well in the previous task are evaluated. Additionally, D-QL necessitates 50 repeated samplings by default for action generation (Wang et al., 2023). This requirement would result in a substantial training overhead for high-dimensional action tasks. Consequently, to ensure a fair comparison, D-QL is configured to allow only 1 sampling, akin to QGPO (Lu et al., 2023).

Table 2 demonstrates the advantages of `LatentDiffuser` become even more pronounced in high-dimensional tasks. Furthermore, a marked decrease in sequence modeling method performance is

Table 2: Adroit results. These tasks have high action dimensionality (24 degrees of freedom)

| Dataset | Environment | CQL | TT | DD | D-QL@1 | TAP | QGPO | HDMI | LD |
|---------|-------------|-----|-----|-----|--------|-----|------|------|-----|
| Human | Pen | 37.5 | 36.4 | $64.1 \pm 9.0$ | $66.0 \pm 8.3$ | $\mathbf{76.5 \pm 8.5}$ | $73.9 \pm 8.6$ | $66.2 \pm 8.8$ | $\mathbf{79.0 \pm 8.1}$ |
| Human | Hammer | $\mathbf{4.4}$ | 0.8 | $1.0 \pm 0.1$ | $1.3 \pm 0.1$ | $1.4 \pm 0.1$ | $1.4 \pm 0.1$ | $1.2 \pm 0.1$ | $\mathbf{4.6 \pm 0.1}$ |
| Human | Door | $\mathbf{9.9}$ | 0.1 | $6.9 \pm 1.2$ | $8.0 \pm 1.2$ | $8.8 \pm 1.2$ | $8.5 \pm 1.2$ | $7.1 \pm 1.1$ | $9.8 \pm 1.0$ |
| Human | Relocate | $\mathbf{0.2}$ | 0.0 | $\mathbf{0.2 \pm 0.1}$ | $\mathbf{0.2 \pm 0.1}$ | $\mathbf{0.2 \pm 0.1}$ | $\mathbf{0.2 \pm 0.1}$ | $0.1 \pm 0.1$ | $\mathbf{0.2 \pm 0.1}$ |
| Cloned | Pen | 39.2 | 11.4 | $47.7 \pm 9.2$ | $49.3 \pm 8.0$ | $57.4 \pm 8.7$ | $54.2 \pm 9.0$ | $48.3 \pm 8.9$ | $\mathbf{60.7 \pm 9.1}$ |
| Cloned | Hammer | 2.1 | 0.5 | $0.9 \pm 0.1$ | $1.1 \pm 0.1$ | $1.2 \pm 0.1$ | $1.1 \pm 0.1$ | $1.0 \pm 0.1$ | $\mathbf{4.2 \pm 0.1}$ |
| Cloned | Door | 0.4 | -0.1 | $9.0 \pm 1.6$ | $10.6 \pm 1.7$ | $\mathbf{11.7 \pm 1.5}$ | $11.2 \pm 1.4$ | $9.3 \pm 1.6$ | $\mathbf{12.0 \pm 1.6}$ |
| Cloned | Relocate | $\mathbf{-0.1}$ | $\mathbf{-0.1}$ | $-0.2 \pm 0.0$ | $-0.2 \pm 0.0$ | $-0.2 \pm 0.0$ | $-0.2 \pm 0.0$ | $\mathbf{-0.1 \pm 0.0}$ | $\mathbf{-0.1 \pm 0.0}$ |
| Expert | Pen | 107.0 | 72.0 | $107.6 \pm 7.6$ | $112.6 \pm 8.1$ | $\mathbf{127.4 \pm 7.7}$ | $119.1 \pm 8.1$ | $109.5 \pm 8.0$ | $\mathbf{131.2 \pm 7.3}$ |
| Expert | Hammer | 86.7 | 15.5 | $106.7 \pm 1.8$ | $114.8 \pm 1.7$ | $\mathbf{127.6 \pm 1.7}$ | $123.2 \pm 1.8$ | $111.8 \pm 1.7$ | $\mathbf{132.5 \pm 1.8}$ |
| Expert | Door | 101.5 | 94.1 | $87.0 \pm 0.8$ | $93.7 \pm 0.8$ | $104.8 \pm 0.8$ | $98.8 \pm 0.8$ | $85.9 \pm 0.9$ | $\mathbf{111.9 \pm 0.8}$ |
| Expert | Relocate | 95.0 | 10.3 | $87.5 \pm 2.8$ | $95.2 \pm 2.8$ | $\mathbf{105.8 \pm 2.7}$ | $102.5 \pm 2.8$ | $91.3 \pm 2.6$ | $\mathbf{109.5 \pm 2.8}$ |
| **Average (w/o expert)** | | 11.7 | 6.1 | 16.2 | 17.1 | 19.6 | 18.79 | 16.6 | **21.3** |
| **Average (w/ expert)** | | 40.3 | 20.1 | 43.2 | 46.1 | **51.9** | 49.5 | 44.3 | **54.6** |

observed. Two primary factors are identified: first, larger action dimensions necessitate tokenization- and autoregression-based techniques (such as TT) to process increasingly lengthy sequences; second, DD and HDMI employ an inverse dynamic model to generate actions independently, while the expansion in action dimension renders the model fitting process more challenging.

## 5.3 LONG-HORIZION CONTINUOUS CONTROL: ANTMAZE

**Baselines** To validate the benefits of latent actions in longer-horizon tasks, an additional comparison is made with hierarchical offline RL methods designed explicitly for long-horizon tasks: CompILE (Kipf et al., 2019), GoFAR (Ma et al., 2022), and HiGoC (Li et al., 2022). Concurrently, CQL and TT are removed due to their inability to perform well in high-dimensional Adroit.

Table 3: AntMaze performance correspond to the mean and standard error over 5 planning seeds.

| Environment | | CompILE | GoFAR | HiGoC | DD | D-QL@1 | TAP | QGPO | HDMI | LD |
|-------------|---------|---------|-------|-------|-----|--------|-----|------|------|-----|
| AntMaze-Play | U-Maze-3 | $41.2 \pm 3.6$ | $38.5 \pm 2.2$ | $31.2 \pm 3.2$ | $73.1 \pm 2.5$ | $52.9 \pm 4.1$ | $82.2 \pm 2.1$ | $59.3 \pm 1.3$ | $\mathbf{86.1 \pm 2.4}$ | $85.4 \pm 1.9$ |
| AntMaze-Diverse | U-Maze-3 | $23.5 \pm 1.8$ | $25.1 \pm 3.1$ | $25.5 \pm 1.6$ | $49.2 \pm 3.1$ | $32.5 \pm 5.9$ | $69.8 \pm 0.5$ | $38.5 \pm 2.6$ | $73.7 \pm 1.1$ | $\mathbf{75.6 \pm 2.1}$ |
| AntMaze-Diverse | Large-2 | - | - | - | $46.8 \pm 4.4$ | - | $69.2 \pm 3.2$ | - | $71.5 \pm 3.5$ | $\mathbf{75.8 \pm 2.0}$ |
| **Single-task Average** | | 32.4 | 31.8 | 28.4 | 56.4 | 39.0 | 73.7 | 45.4 | **77.1** | **78.9** |
| MultiAnt-Diverse | Large-2 | - | - | - | $45.2 \pm 4.9$ | - | $\mathbf{71.6 \pm 3.3}$ | - | $73.6 \pm 3.8$ | $73.3 \pm 2.6$ |
| **Multi-task Average** | | - | - | - | 45.2 | - | **71.6** | - | 73.6 | 73.3 |

Table 3 highlights that sequence modeling-based hierarchical methods significantly surpass RL-based approaches. Moreover, `LatentDiffuser` demonstrates performance comparable to two-stage techniques such as TAP and HDMI through end-to-end training.

## 6 CONCLUSIONS

In this work, we present a novel approach, `LatentDiffuser`, for tackling temporal-extended offline tasks, addressing the limitations of previous state-of-the-art offline reinforcement learning methods and conditional generation models in handling high-dimensional, long-horizon tasks. `LatentDiffuser` is capable of end-to-end learning for both representation of and planning with latent action, delivering a unified, comprehensive solution for offline decision-making and control. Numerical results on *Gym locomotion control*, *Adroit*, and *AntMaze*, demonstrate the effectiveness of `LatentDiffuser` in comparison with existing hierarchical- and planning-based offline methods. The performance gains are particularly noticeable in high-dimensional and long-horizon tasks, illustrating the advantages of `LatentDiffuser` in addressing these challenging scenarios.

### ACKNOWLEDGMENTS

We extend our heartfelt gratitude to Professor Hongyuan Zha for his enlightening discussions. This work was supported in part by Postdoctoral Science Foundation of China (2022M723039).

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

# Supplementary Material

## Table of Contents

## A  LIMITATIONS AND SOCIETAL IMPACT

**Limitations**  `LatentDiffuser`, analogous to other diffusion-based methods for offline decision-making, exhibits a protracted inference time owing to the iterative nature of the sampling process. This challenge could be alleviated through the adoption of approaches that enable accelerated sampling (Lu et al., 2022a;b) or by distilling these diffusion models into alternative methods necessitating fewer sampling iterations (Song et al., 2023). Additionally, similar with TAP (Jiang et al., 2023), empirical findings from continuous control featuring deterministic dynamics indicate that `LatentDiffuser` can manage epistemic uncertainty. However, the efficacy of `LatentDiffuser` in addressing tasks characterized by stochastic dynamics without modifications remains unascertained. Furthermore, a deficiency in our methodology is the requirement for both the latent steps $L$ and planning horizon $H$ for the latent action to remain constant. We hope that facilitating adaptive variation of these hyperparameters may enhance performance.

**Societal Impact** Similar with other deep generative modeling techniques, the energy-guided diffusion sampling employed in this paper possesses the potential to generate harmful content and may perpetuate and exacerbate pre-existing undesirable biases present in the offline dataset.

## B PRELIMINARIES

### B.1 OFFLINE REINFORCEMENT LEARNING

In general, reinforcement learning (RL) represents the problem of sequential decision-making through a Markov Decision Process $\mathcal{M} = (\mathcal{S}, \mathcal{A}, \mathcal{P}, r, \gamma)$, encompassing a state space $\mathcal{S}$ and an action space $\mathcal{A}$. Given states $s, s' \in \mathcal{S}$ and an action $a \in \mathcal{A}$, the transition probability function is expressed as $\mathcal{P}(s' \mid s, a) : \mathcal{S} \times \mathcal{A} \times \mathcal{S} \to [0, 1]$ and the reward function is defined by $r(s, a, s') : \mathcal{S} \times \mathcal{A} \times \mathcal{S} \to \mathbb{R}$. The discount factor is denoted as $\gamma \in (0, 1]$.

The policy is represented as $\pi : \mathcal{S} \times \mathcal{A} \to [0, 1]$, indicating the probability of taking action $a$ in state $s$ as $\pi(a \mid s)$. For the timestep $t \in [1, T]$, the cumulative discounted reward, also referred to as *reward-to-go*, is identified by $R_t = \sum_{t'=t}^{T} \gamma^{t'-t} r_{t'}$. The principal objective of *online* RL is to determine a policy $\pi$ that maximizes $J = \mathbb{E}_{a_t \sim \pi(\cdot|s_t), s_{t+1} \sim \mathcal{P}(\cdot|s_t, a_t)} [\sum_{t=1}^{T} \gamma^{t-1} r_t(s_t, a_t, s_{t+1})]$ via learning from transitions $(s, a, r, s')$ during environment interaction (Sutton & Barto, 2018).

Conversely, in offline RL, a static dataset $\mathcal{D}$ is employed, which has been collected through a behavior policy $\pi_\mu$, for acquiring a policy $\pi$ that optimizes $J$ for subsequent application in the interactive environment. The behavior policy $\pi_\mu$ can either constitute a single policy or an amalgamation of various policies; however, it remains inaccessible. The acquisition of data is presumed to be trajectory-wise, as represented by $\mathcal{D} = \{\tau_i\}_{i=1}^{D}$, where $\tau = \{(s_i, a_i, r_i, s'_i)\}_{i=1}^{T}$.

### B.2 DIFFUSION PROBABILISTIC MODELS

This section will provide an introduction to the diffusion probabilistic model within the context of the `LatentDiffuser`. Diffusion probabilistic models (Sohl-Dickstein et al., 2015; Ho et al., 2020), constitute a likelihood-based generative framework that facilitates learning data distributions $q(z)$ from the offline datasets expressed as $\mathcal{D} := \{z^i\}$, wherein the index $i$ denotes a specific sample within the dataset (Song, 2021), and $z^i$ is the latent actions encoded by the pre-trained encoder $q_\phi$. A core concept within diffusion probabilistic models lies in the representation of the (Stein) score function (Liu et al., 2016), which does not necessitate a tractable normalizing constant (also referred to as the *partition function*).

The discrete-time generation procedure encompasses a designed forward noising (or diffusion) process $q(z_{k+1}|z_k) := \mathcal{N}(z_{k+1}; \sqrt{\tilde{\alpha}_k} z_k, (1 - \tilde{\alpha}_k)\mathbf{I})$ at (forward) diffusion timestep $k$. The forward process coupled with a learnable, reverse denoising (or diffusion) process $p_\theta(z_{k-1}|z_k) := \mathcal{N}(z_{k-1}|\mu_\theta(z_k, k), \Sigma_k)$ at (backward) diffusion timestep $k$. $\mathcal{N}(\mu, \Sigma)$ signifies a Gaussian distribution characterized by mean $\mu$ and variance $\Sigma$, $\alpha_k \in \mathbb{R}$ establishes the variance schedule. In order to ensure consistency with the main text notation, we denote $\alpha_k := \sqrt{\tilde{\alpha}_k}$ and $\sigma_k := \sqrt{1 - \tilde{\alpha}_k}$. $z_0 := z$ corresponds to a sample in $\mathcal{D}$, $z_1, z_2, \ldots, z_{K-1}$ signifies the latent variables or the noised latent actions, and $z_K \sim \mathcal{N}(\mathbf{0}, \mathbf{I})$ for judiciously selected $\tilde{\alpha}_k$ values and a sufficiently extensive $K$.

Commencing with Gaussian noise, samples undergo iterative generation via a sequence of denoising steps. An optimizable and tractable variational lower-bound on $\log p_\theta$ serves to train the denoising operator, with a simplified surrogate loss proposed in (Ho et al., 2020):

$$\mathcal{L}_{\text{denoise}}(\theta) := \mathbb{E}_{k \sim [1, K], z_0 \sim q, \epsilon \sim \mathcal{N}(\mathbf{0}, \mathbf{I})} \left[ \|\epsilon - \epsilon_\theta(z_k, k)\|^2 \right]. \tag{8}$$

The predicted noise $\epsilon_\theta(z_k, k)$, parameterized through a deep neural network, emulates the noise $\epsilon \sim \mathcal{N}(0, I)$ integrated with the dataset sample $z_0$ yielding noisy $z_k$ in the noising process.

**Conditional Diffusion Probabilistic Models** Intriguingly, the conditional distribution $q(z|y(z))$ facilitates sample generation under the condition $y(z)$. Within the context of this paper, $y(z)$ is instantiated as the initial state $s_1$. The equivalence between diffusion probabilistic models and score-matching (Song et al., 2021) reveals that $\epsilon_\theta(z_k, k) \propto \nabla_{z_k} \log p(z_k)$, giving rise to two categorically

equivalent methodologies for conditional sampling with diffusion probabilistic models: classifier-guided (Nichol & Dhariwal, 2021), and classifier-free (Ho & Salimans, 2022) techniques employed in our work. The latter method modifies the preliminary training configuration, learning both a conditional $\epsilon_\theta(z_k, s_1, k)$ and an unconditional $\epsilon_\theta(z_k, k)$ model for noise. Unconditional noise manifests as conditional noise $\epsilon_\theta(z_k, \varnothing, k)$, with a placeholder $\varnothing$ replacing $s_1$. When $y(z) = \varnothing$, the entries of $e$ are zeroed out. The perturbed noise $\tilde{\epsilon}_k := \epsilon_\theta(z_k, k) + \omega(\epsilon_\theta(z_k, s_1, k) - \epsilon_\theta(z_k, k))$ is subsequently employed to generate samples. Additionally, we adopt low-temperature sampling in the denoising process to ensure higher quality latent actions (Ajay et al., 2023). Concretely, we compute $\mu_{k-1}$ and $\Sigma_{k-1}$ from the previous noised latent actions $z_{k-1}$ and perturbed noise $\tilde{\epsilon}_{k-1}$, and subsequently sample $z_{k-1} \sim \mathcal{N}(\mu_{k-1}, \alpha\Sigma_{k-1})$ with the variance scaled by $\alpha \in [0, 1)$.

## C MISSING RELATED WORK

### C.1 MODEL-BASED REINFORCEMENT LEARNING

`LatentDiffuser` is incorporated into a research trajectory focused on model-based reinforcement learning (RL) (Sutton, 1990; Janner et al., 2019; Schrittwieser et al., 2020; Lu et al., 2021; Eysenbach et al., 2022; Suh et al., 2023), as it makes decisions by forecasting future outcomes. These approaches frequently employ predictions in the raw Markov Decision Process (MDP), which entails that models accept the current raw state and action as input, outputting probability distributions encompassing subsequent states and rewards. Hafner et al. (2019), Ozair et al. (2021), Hafner et al. (2021), Hafner et al. (2023), and Chitnis et al. (2023) proposed to acquiring a latent state space in conjunction with a dynamics function. Contrarily, in their cases, the action space accessible to the planner remains identical to that of the raw MDP, and the execution of the plan maintains its connection to the original temporal structure of the environment.

### C.2 ACTION REPRESENTATION LEARNING.

The concept of learning a representation for actions and conducting RL within a latent action space has been investigated in the context of model-free RL (Merel et al., 2019; Allshire et al., 2021; Zhou et al., 2021; Chen et al., 2022; Peng et al., 2022; Dadashi et al., 2022). In contrast to `LatentDiffuser`, where the latent action space is utilized to promote efficacy and robustness in planning, the motivations for obtaining a latent action space in model-free approaches vary, yet the underlying objective centers on providing policy constraints. For instance, Merel et al. (2019) and Peng et al. (2022) implement this concept for humanoid control to ensure the derived policies resemble low-level human demonstration behavior, thus being classified as natural. Zhou et al. (2021) and Chen et al. (2022) employ latent actions to prevent out-of-distribution (OOD) actions within the offline RL framework. Dadashi et al. (2022) proposes adopting a discrete latent action space to facilitate the application of methods designed explicitly for discrete action spaces to continuous cases. In teleoperation literature, Karamcheti et al. (2021) and Losey et al. (2022) embed high-dimensional robotic actions into lower-dimensional, human-controllable latent actions.

Additionally, several works focus on learning action representations for improved planning efficiency. Wang et al. (2020) and Yang et al. (2021) learn action representations for on-the-fly learning, applicable to black-box optimization and path planning scenarios. Despite high-level similarities, these papers assume prior knowledge of environment dynamics. TAP (Jiang et al., 2023) extends this framework into the offline RL domain, where the actual environmental dynamics remain undetermined, necessitating joint learning of the dynamics model and the representation of latent action. Nevertheless, TAP is constrained to a discrete latent action space and demands costly additional planning. `LatentDiffuser` can achieve representation learning and planning for continuous latent actions by leveraging the latent diffusion model in an end-to-end manner.

### C.3 OFFLINE REINFORCEMENT LEARNING.

`LatentDiffuser` is devised for the offline RL (Ernst et al., 2005; Levine et al., 2020), precluding the utilization of online experiences for policy improvement. A principal hurdle in offline RL involves preventing out-of-distribution (OOD) actions selection by the learned policy to circumvent value function and model inaccuracies exploitation. Conservatism (Kumar et al., 2020; Kidambi et al.,

2020b; Fujimoto & Gu, 2021; Lu et al., 2022c; Kostrikov et al., 2022) is proposed as a standard solution for this challenge. `LatentDiffuser` inherently prevents OOD actions via planning in a learned latent action space.

Pursuing adherence to a potentially diverse behavior policy, recent works have identified diffusion models as powerful generative tools, which generally surpass preceding generative approaches such as Gaussian distribution (Peng et al., 2019; Wang et al., 2020b; Nair et al., 2020) and Variational Autoencoders (VAEs) (Fujimoto et al., 2019; Wang et al., 2021) concerning behavior modeling. Different methods adopt distinct strategies for action generation, maximizing the learned $Q$-functions. Diffusion-QL (Wang et al., 2023) monitors gradients from behavior diffusion policy-derived actions to guide generated actions towards higher $Q$-value regions. SfBC (Chen et al., 2023b) and Diffusion-QL employ a similar idea, whereby resampling actions from multiple behavioral action candidates occur, with predicted $Q$-values serving as sampling weights. Ada et al. (2023) incorporates a state reconstruction loss-based regularization term within the diffusion-based policy training, consequently bolstering generalization capabilities for OOD states. Alternative works (Goo & Niekum, 2022; Pearce et al., 2023; Block et al., 2023; Suh et al., 2023) solely deploy diffusion models for behavior cloning or planning, rendering $Q$-value maximization unnecessary.

Contrary to the works above aligned with the RL paradigm, `LatentDiffuser` addresses offline RL challenges through sequence modeling (refer to the subsequent section). Compared to RL-based offline methodologies, sequence modeling offers benefits regarding temporally extended and sparse or delayed reward tasks.

### C.4    REINFORCEMENT LEARNING AS SEQUENCE MODELING

`LatentDiffuser` stems from an emerging body of research that conceptualizes RL as a sequential modeling problem (Bhargava et al., 2023). Depending on the model skeleton, this literature may be classified into two primary categories. The first category comprises models that leverage a GPT-2 (Radford et al., 2019) style Transformer architecture (Vaswani et al., 2017), also referred to as causal transformers, for the autoregressive modeling of states, actions, rewards, and returns, ultimately converting predictive capabilities into policy. Examples include Decision Transformer (DT, Chen et al. 2021) and Zheng et al. (2022), which apply an Upside Down RL technique (Schmidhuber, 2019) under both offline and online RL settings, and Trajectory Transformer (TT, Janner et al. 2021), which employs planning to obtain optimal trajectories maximizing return. Chen et al. (2023a) introduced a non-autoregressive planning algorithm based on energy minimization, while Jia et al. (2023) enhanced generalization ability for unseen tasks through refined in-context example design. Lastly, Wu et al. (2023) addresses trajectory stitching challenges by adjusting the history length employed in DT.

The second category features models based on a score-based diffusion process for non-autoregressive modeling of state and action trajectories. Different methods select various conditional samplers to generate actions that maximize the return. Diffuser (Janner et al., 2022) emulates the classifier-guidance methodology (Nichol & Dhariwal, 2021) and employing guidance methods as delineated in § F.1. Alternatively, Decision Diffuser (Ajay et al., 2023) and its derivatives (Li et al., 2023; Hu et al., 2023) explore classifier-free guidance (Ho & Salimans, 2022). Extensions of this concept to multi-task settings are presented by He et al. (2023) and Ni et al. (2023), while Liang et al. (2023) utilizes the diffusion model as a sample generator for unseen tasks, thus improving generalization capabilities. `LatentDiffuser` offers a more efficient planning solution to enable these sequential modeling algorithms to navigate complex action spaces effectively.

### C.5    CONTROLLABLE SAMPLING WITH GENERATIVE MODELS

`LatentDiffuser` produces trajectories corresponding to optimal policies by employing controllable sampling within diffusion models. Current methods for facilitating controllable generation in diffusion models primarily emphasize conditional guidance. Such approaches leverage a pretrained diffusion model for the definition of the prior distribution $q(x)$ and strive to obtain samples from $q(x) \exp(-\beta \mathcal{E}(x))$. Graikos et al. (2022) introduces a training-free sampling technique, which finds application in approximating solutions for traveling salesman problems. Poole et al. (2023) capitalizes on a pretrained 2D diffusion model and optimizes 3D parameters for generating 3D shapes. Kawar et al. (2022) and Chung et al. (2023) exploit pretrained diffusion models for addressing linear and specific non-linear inverse challenges, such as image restoration, deblurring, and denoising. Dif-

fuser (Janner et al., 2022) and Decision Diffuser (Ajay et al., 2023) use pretrained diffusion models to solve the offline RL problem. Zhao et al. (2022) and Bao et al. (2023) employ human-crafted intermediate energy guidance for tasks including image-to-image translation and inverse molecular design. In a more recent development, Lu et al. (2023) presents a comprehensive framework for incorporating human control within the sampling process of diffusion models.

**Remark** Recently, Venkatraman (2023) proposed a novel algorithm called LDCQ, which is very similar to `LatentDiffuser`. Specifically, LDCQ also introduces a latent diffusion model to learn a latent action space. Unlike the latent action used by LatentDiffuser, the learned latent action space in LDCQ belongs to the skill space, similar to what is described in Appendix F.2. Additionally, LDCQ does not perform planning within the learned skill space but instead uses model-free TD-learning methods to choose the optimal skill at each timestep and obtains the final action using a decoder. In summary, it is quite a coincidence that LDCQ and `LatentDiffuser` belong to two orthogonal approaches to utilizing latent diffusion models in offline RL. The former still adopts the RL framework to model the offline RL problems, while LatentDiffuser approaches the problem from a conditional generative perspective.

## D  MISSING RESULTS AND ANALYSES

### D.1  PROOF-OF-CONCEPT EXAMPLE: MAZE-2D-OPEN

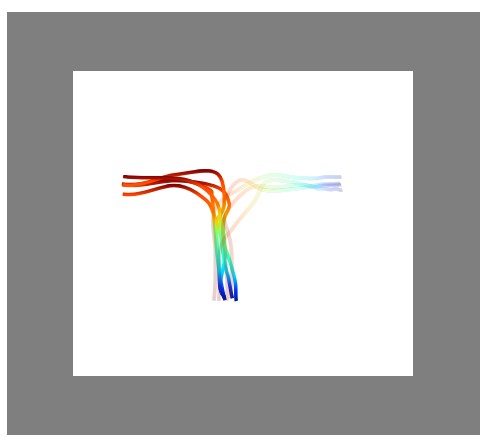
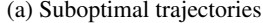
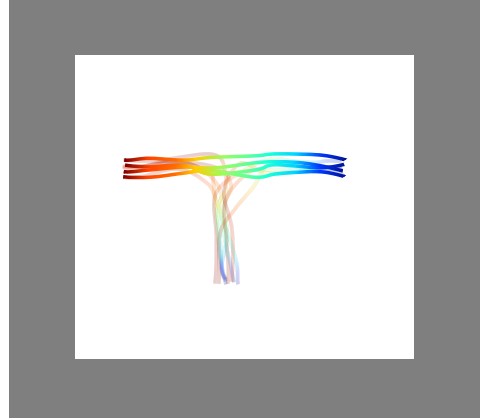

(a) Suboptimal trajectories.          (b) Stitched near optimal trajectories.

Figure 3: Proof-of-Concept example. We demonstrate the importance of planning through an experiment designed by Decison Diffuser (Ajay et al., 2023, DD; Appendix A.1). The diffusion model achieves trajectory stitching, a process essential for handling a large number of suboptimal trajectories, through implicit planning.

To demonstrate the importance of planning, we designed an experiment same as Ajay et al. (2023, Appendix A.1). Specifically, most tasks' offline datasets contain a large number of suboptimal trajectories. To learn better policies rather than just simple behavior cloning, trajectory stitching is one of the essential abilities algorithms must possess. To validate whether `LatentDiffuser` can achieve trajectory stitching through implicit planning, we adopted the experimental setup same as Ajay et al. (2023, Appendix A.1). In the `maze-2D-open` environment, the objective is to navigate towards the target area situated on the right side, with the reward being the negative distance to this target area. The training dataset is composed of 500 trajectories originating from the left side and terminating at the bottom side, as well as 500 trajectories starting from the bottom side and ending at the right side. Each trajectory is constrained to a maximum length of 50. At test time, the agent begins on the left side and aims to reach the right side as efficiently as possible. As demonstrated in Figure 3 and consistent with the findings of Ajay et al. (2023, Appendix A.1), the `LatentDiffuser` can effectively stitch trajectories from the training dataset to produce trajectories that traverse from the left side to the right side in (near) straight lines.

## D.2 MAIN RESULTS

The performance in Gym locomotion control in terms of normalized average returns of all baselines are shown in Table 4.

Table 4: The performance in Gym locomotion control in terms of normalized average returns of all baselines. Results correspond to the mean and standard error over 5 planning seeds.

| Dataset | Environment | CQL | IQL | DT | TT | MoReL | Diffuser |
|---------|-------------|-----|-----|-----|-----|-------|----------|
| Med-Expert | HalfCheetah | 91.6 | 86.7 | 86.8 | **95** | 53.3 | 79.8 |
| Med-Expert | Hopper | 105.4 | 91.5 | 107.6 | **110.0** | **108.7** | 107.2 |
| Med-Expert | Walker2d | **108.8** | **109.6** | **108.1** | 101.9 | 95.6 | **108.4** |
| Medium | HalfCheetah | 44.0 | 47.4 | 42.6 | 46.9 | 42.1 | 44.2 |
| Medium | Hopper | 58.5 | 66.3 | 67.6 | 61.1 | **95.4** | 58.5 |
| Medium | Walker2d | 72.5 | 78.3 | 74.0 | 79 | 77.8 | 79.7 |
| Med-Replay | HalfCheetah | **45.5** | **44.2** | 36.6 | 41.9 | 40.2 | 42.2 |
| Med-Replay | Hopper | 95 | 94.7 | 82.7 | 91.5 | 93.6 | 96.8 |
| Med-Replay | Walker2d | 77.2 | 73.9 | 66.6 | **82.6** | 49.8 | 61.2 |
| **Average** | | 77.6 | 77 | 74.7 | 78.9 | 72.9 | 75.3 |
| **Dataset** | **Environment** | **DD** | **D-QL** | **TAP** | **QGPO** | **HDMI** | **LatentDiffuser** |
| Med-Expert | HalfCheetah | 90.6±1.3 | 96.8±0.3 | 91.8 ± 0.8 | 93.5±0.3 | 92.1±1.4 | 95.2±0.2 |
| Med-Expert | Hopper | **111.8±1.8** | 111.1±1.3 | 105.5 ± 1.7 | 108.0±2.5 | 113.5±0.9 | 112.9±0.3 |
| Med-Expert | Walker2d | **108.8±1.7** | 110.1±0.3 | 107.4 ± 0.9 | 110.7 ± 0.6 | 107.9±1.2 | 111.3±0.2 |
| Medium | HalfCheetah | 49.1±1.0 | 51.1±0.5 | 45.0 ± 0.1 | 54.1 ± 0.4 | 48.0±0.9 | 53.6±0.4 |
| Medium | Hopper | 79.3±3.6 | 90.5±4.6 | 63.4 ± 1.4 | 98.0 ± 2.6 | 76.4±2.6 | 98.5±0.7 |
| Medium | Walker2d | 82.5±1.4 | 87.0±0.9 | 64.9 ± 2.1 | 86.0 ± 0.7 | 79.9±1.8 | 86.3±0.9 |
| Med-Replay | HalfCheetah | 39.3±4.1 | 47.8±0.3 | 40.8 ± 0.6 | 47.6 ± 1.4 | 44.9±2.0 | 47.3±1.2 |
| Med-Replay | Hopper | 100±0.7 | 101.3±0.6 | 87.3 ± 2.3 | 96.9 ± 2.6 | 99.6±1.5 | 100.4±0.5 |
| Med-Replay | Walker2d | 75±4.3 | 95.5±1.5 | 66.8 ± 3.1 | 84.4 ± 4.1 | 80.7±2.1 | 82.6 ± 2.1 |
| **Average** | | 81.8 | **88.0** | 82.5 | 86.6 | 82.6 | **87.5** |

This section will then delve into a more detailed analysis of the performance differences among different baselines across various tasks. To provide an intuitive comparison of different algorithms, we classify them from three perspectives — planning, hierarchy, and generative — according to the classification method shown in Table 5. Firstly, the Gym locomotion task has a long horizon, dense rewards, and low action dimensions, making it a baseline test task for offline RL. The results from Table 4 show that generative methods based on diffusion models generally perform better. The community currently attributes this to diffusion models' more powerful representation capabilities in modeling more complex policies or environmental models. However, `LatentDiffuser` does not demonstrate its advantages well in the low-dimensional action space. Although `LatentDiffuser` approaches the SOTA performance on this task, it is mainly due to a better diffusion sampling method, which is supported by the solid performance of the QGPO method. Due to dense rewards, planning and hierarchy-based methods, such as TAP and HDMI, have not achieved the best results.

Secondly, the Adroit task is characterized by a high-dimensional action space. This leads to the best performance for TAP and LatentDiffuser (see Table 2), two methods based on latent action, which experimentally verify the effectiveness of latent action. Additionally, generative methods based on diffusion models generally exhibit better performance. However, due to the shorter horizon of the Adroit task, the HDMI method, which is based on planning and hierarchy, does not achieve the best performance.

Lastly, the AntMaze task has a longer horizon and very sparse rewards. This allows latent action ample room for improvement (see Table 3). Moreover, methods based on planning and hierarchy also achieve good results, such as HDMI. In this task, non-generative methods based on planning and hierarchy, such as ComPILE and GoFAR, approach the performance of generative methods without planning and hierarchy (D-QL).

**The Performance Gap Between TAP and `LatentDiffuser`** For TAP and LatentDiffuser, the performance gap between them on the expert dataset is smaller than on other datasets in Adroit and Gym locomotion tasks. We analyzed that the primary source of this performance gap comes from

the proportion of suboptimal trajectories in the dataset. In non-expert datasets, the proportion of suboptimal trajectories is more significant. To learn the optimal policy from the dataset, the algorithm needs to have the "trajectory stitch" ability, i.e., to splice segments of suboptimal trajectories to form an optimal trajectory.

On the one hand, most of the current offline RL methods are based on a dynamic programming framework to learn a Q function. However, these methods require the Q function to have Bellman completeness to achieve good performance. Designing a function class with Bellman completeness is very challenging (Zhou et al., 2023). On the other hand, Ajay et al. (2023, Appendix A.1) has found that generative methods based on diffusion models possess implicit dynamic programming capabilities. These methods use the powerful representation ability of diffusion models to bypass Bellman completeness and achieve the "trajectory stitch" ability. This allows them to perform well in datasets with more suboptimal trajectories.

`LatentDiffuser` is a generative method based on a diffusion model, while TAP is not. This leads to a more significant performance gap between the two on non-expert datasets. In expert datasets, however, `LatentDiffuser`'s advantage cannot be demonstrated.

## E  IMPLEMENTATION AND TRAINING DETAILS

In the following subsection, we delineate hyperparameter configurations and training methodologies employed in numeric experiments for both baseline models and the proposed `LatentDiffuser`. Additionally, we supply references for performance metrics of prior evaluations conducted on standardized tasks concerning baseline models.

Each task undergoes assessment with a total of 5 distinct training seeds, evaluated over a span of 20 episodes. Adhering to the established evaluation protocols of TT (Janner et al., 2021) and IQL (Kostrikov et al., 2022), the dataset versions employed for locomotion control experiments are defined as $v2$, whereas $v0$ versions are utilized for remaining tasks.

### E.1  BASELINE DETAILS

Before discussing the specific baseline implementation details, we first made a simple comparison of all baselines in the 3 tasks from 3 perspectives: whether *planning* is introduced, whether it contains *hierarchical* structure, and whether *generative* learning is introduced, as shown in Table 5.

Table 5: Comparison of different baselines at three levels. ● means inclusive, ○ means exclusive, and ◐ means a cheaper approximation.

|  | ComPILE | CQL | IQL | D-QL | D-QL@1 | QGPO | Diffuser | DD |
|---|---|---|---|---|---|---|---|---|
| **Planning** | ○ | ○ | ○ | ○ | ○ | ○ | ◐ | ◐ |
| **Hierarchy** | ● | ○ | ○ | ○ | ○ | ○ | ○ | ○ |
| **Generative** | ○ | ○ | ○ | ● | ● | ● | ● | ● |
|  | DT | TT | MoReL | HiGoC | GoFAR | TAP | HDMI | LatentDiffuser |
| **Planning** | ○ | ● | ● | ● | ● | ● | ◐ | ◐ |
| **Hierarchy** | ○ | ○ | ○ | ● | ● | ● | ● | ● |
| **Generative** | ● | ● | ○ | ○ | ○ | ● | ● | ● |

### E.1.1  GYM LOCOMOTION CONTROL

- The results of CQL in Table 1 and Table 4 is reported in (Kostrikov et al., 2022, Table 1);
- The results of IQL in Table 4 is reported in (Kostrikov et al., 2022, Table 1);
- The results of DT in Table 4 is reported in (Chen et al., 2021, Table 2);
- The results of TT in Table 1 and Table 4 is reported in (Janner et al., 2021, Table 1);
- The results of MoReL in Table 4 is reported in (Kidambi et al., 2020, Table 2);
- The results of Diffuser in Table 4 is reported in the (Janner et al., 2022, Table 2);

- The results of DD in Table 1 and Table 4 is reported in the (Ajay et al., 2023, Table 1).

- The results of D-QL in Table 1 and Table 4 is reported in the (Wang et al., 2023, Table 1).

- The results of TAP in Table 1 and Table 4 is reported in the (Jiang et al., 2023, Table 1).

- The results of QGPO in Table 1 and Table 4 is reported in the (Lu et al., 2023, Table 2).

- The results of HDMI in Table 1 and Table 4 is reported in the (Li et al., 2023, Table 3).

### E.1.2 ADROIT ENVIRONMENT

- The results of CQL in Table 2 is reported in (Kostrikov et al., 2022, Table 1);

- The results of DD in Table 2 is generated by using the offical repository[3] from the original paper (Ajay et al., 2023) with default hyperparameters.

- The results of D-QL@1 in Table 2 is generated by using the official repository[4] from the original paper (Wang et al., 2023) with default hyperparameters.

- The results of QGPO in Table 2 is generated by using the offcial repository[5] from the original paper (Lu et al., 2023) with default hyperparameters.

- The results of HDMI in Table 2 is generated by re-implementing the algorithm from the original paper (Li et al., 2023) with default hyperparameters.

- The results of TT in Table 2 is reported in (Janner et al., 2021, Table 1);

- The results of TAP in Table 2 is reported in the (Jiang et al., 2023, Table 1).

It is essential to mention that the D-QL employs a resampling procedure for assessment purposes. To be more precise, during evaluation, the acquired policy initially produces 50 distinct action candidates, subsequently selecting a single action possessing the highest $Q$-value for execution. We empirically find that that this strategy is critical for achieving satisfactory performance in Adroit tasks. Nonetheless, the technique poses challenges in accurately representing the quality of initially sampled actions prior to the resampling procedure. Additionally, due to the high dimensionality of the action, it incurs considerable computational overhead. As a result, the resampling process has been eliminated from the evaluation, utilizing a single action candidate (referred to as D-QL@1) akin to QGPO (Lu et al., 2023).

### E.1.3 ANTMAZE ENVIRONMENT

- The results of ComPILE, GoFAR, DD and HDMI in Table 3 is reported in (Li et al., 2023, Table 2);

- The results of HiGoC in Table 3 is generated by re-implementing HiGoC (Li et al., 2022) based on CQL[6] and cVAE[7], and tune over the two hyparameters, learning rate $\in [3e - 4, 1e - 3]$ and the contribution of KL regularization $\in [0.05, 0.2]$.

- The results of D-QL@1 in Table 3 is generated by using the official repository from the original paper (Wang et al., 2023) with default hyperparameters.

- The results of TAP in Table 3 is generated by using the official repository[8] from the original paper (Wang et al., 2023) with default hyperparameters.

- The results of QGPO in Table 3 is generated by using the official repository from the original paper (Wang et al., 2023) with default hyperparameters.

---

[3] https://github.com/anuragajay/decision-diffuser/tree/main/code.
[4] https://github.com/Zhendong-Wang/Diffusion-Policies-for-Offline-RL.
[5] https://github.com/thu-ml/CEP-energy-guided-diffusion.
[6] https://github.com/aviralkumar2907/CQL.
[7] https://github.com/timbmg/VAE-CVAE-MNIST.
[8] http://github.com/ZhengyaoJiang/latentplan.

### E.2 Implementation Details

The forthcoming release of the complete source code will be subject to the Creative Commons Attribution 4.0 License (CC BY), with the exception of the gym locomotion control, Adroit, and AntMaze datasets, which will retain their respective licensing arrangements. The computational infrastructure consists of dual servers, each possessing 256 GB of system memory, as well as a pair of NVIDIA GeForce RTX 3090 graphics processing units equipped with 24 GB of video memory.

#### E.2.1 Representation Learning for Latent Action

**Encoder and Decoder.** The score-based prior model is significantly influenced by the architecture of the bottleneck, encompassing both the encoder and decoder, and subsequently impacts its application for planning. Adhering to the Decision Transformer Chen et al. (2021) and Trajectory Transformer (Janner et al., 2021), TAP (Jiang et al., 2023) employs a GPT-2-style transformer incorporating *causal masking* within its encoder and decoder. Consequently, information from future tokens does not propagate backward to their preceding counterparts. However, this conventional design remains prevalent in sequence modeling without guaranteeing optimality. For instance, one could invert the masking order in the decoder, thus rendering the planning goal-based.

A detailed examination of the autoregressive (causal) versus simultaneous generation of optimal action sequences can be found in (Janner et al., 2022, §3.1). The discourse presented in Janner et al. (2022) remains germane to the context of latent action space. In particular, it is reasonable to assume that latent action generation adheres to causality, whereby subsequent latent actions are contingent upon previous and current latent actions. However, decision-making or optimal control may exhibit anti-causality, as the subsequent latent action may rely on future information, such as future rewards. In general RL scenarios, the dependence on future information originates from the presumption of future optimality, intending to develop a dynamic programming recursion. This notion is reflected by the future optimality variables $\mathcal{O}_{t:T}$ present in the action distribution $\log p\left(\mathbf{a}_t \mid \mathbf{s}_t, \mathcal{O}_{t:T}\right)$ (Levine, 2018). The aforementioned analysis lends support to the "diffusion-sampling-as-planning" framework. As a result, causal masking is eliminated from the GPT-2 style encoder and state decoder design within the `LatentDiffuser`.

Additionally, the action decoder is represented through the implementation of a 2-layered MLP, encompassing 512 hidden units and ReLU activation functions, effectively constituting an inverse dynamics model. Concurrently, a 3-layered MLP, containing 1024 hidden units and ReLU activation functions, represents the reward and return decoder. The action decoder is trained employing the Adam optimizer, featuring a learning rate of $2e-4$ and batch size of 32 across $2e6$ training steps. The reward and return decoder are also trained utilizing the Adam optimizer, however, with a learning rate of $2e-4$ and batch size of 64 spanning $1e6$ training steps.

**Score-based Prior.** Consistent with DD (Ajay et al., 2023), the score-based prior model is characterized by a temporal U-Net[9] (Janner et al., 2022) architecture encompassing a series of 6 recurrent residual blocks. Within each block, two sequential temporal convolutions are implemented, succeeded by group normalization (Wu & He, 2018), culminating in the application of the Mish activation function (Misra, 2019). Distinct 2-layer MLPs, each possessing 256 hidden units and the Mish activation function, yield 128-dimensional timestep and condition embeddings, which are concatenated and added to the first temporal convolution's activations within each block. We employ the Adam optimization algorithm, utilizing a learning rate of $2 \times 10^{-4}$, a batch size of 32, and performing $2 \times 10^6$ training iterations. The probability, denoted by $p$, of excluding conditioning information $s_1$ is set to 0.25, and $K = 100$ diffusion steps are executed.

#### E.2.2 Planning with Energy-guided Sampling

The energy guidance model is formulated as a 4-layer MLP containing 256 hidden units and leverages SiLU activation functions (Hendrycks & Gimpel, 2016). It undergoes training for $1 \times 10^6$ gradient-based steps, implementing the Adam optimizer with a learning rate of $3 \times 10^{-4}$ and a batch size of 256. For gym locomotion control and Adroit tasks, the dimension of the latent actions set, $M$, is

---

[9]https://github.com/jannerm/diffuser.

established at 16, whereas for AntMaze tasks, it is fixed at 32. In all tasks, however, $\beta$ is set to 3. The planning horizon, $H$, is set to 40 for gym locomotion control and Adroit tasks, and 100 for AntMaze tasks. The guidance scale, $w$, is selected from the range $\{1.2, 1.4, 1.6, 1.8\}$, although the specific value depends on the task. A low temperature sampling parameter, $\alpha$, is designated as 0.5, and the context length, $C$, is assigned a value of 20.

## F  MODELING SELECTION

In this section, we shall explore the alterations elicited by diverse problem modeling approaches upon the `LatentDiffuser` framework. Specifically, by examining various representations of latent action, we investigate the performance of the `LatentDiffuser` within both the *raw* action space and *skill* space. Subsequently, by comparing it against existing works, we distill some key observations of distinct diffusion sampling techniques on the efficacy of planning.

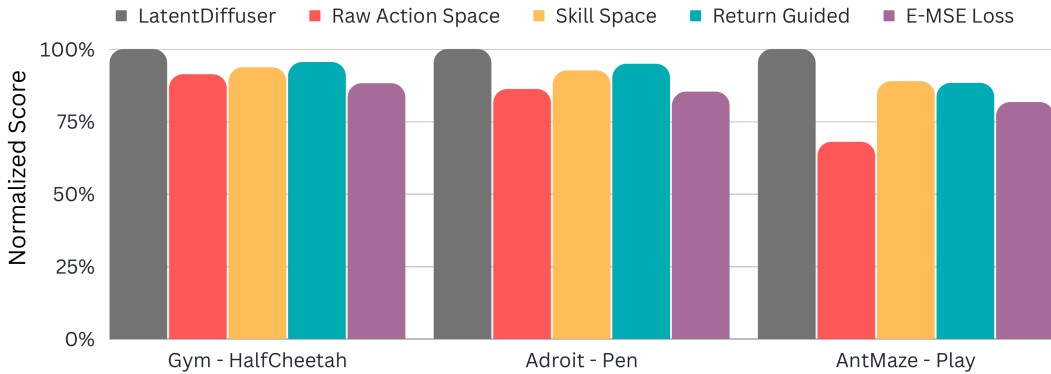

Figure 4: Results of different modeling selections, where the height of the bar is the mean normalised scores on different tasks.

### F.1  PLANNING IN THE RAW ACTION SPACE

The `LatentDiffuser` framework can be equally implemented within the raw action space; in this instance, merely substitute the latent diffusion model with any diffusion model for estimating raw trajectory distributions, such as those employed within the Diffuser (Janner et al., 2022) and Decision Diffuser (Ajay et al., 2023). To be precise, we can deduce the following theorem, akin to Theorem 1, within the raw action space:

---

**Theorem 2** (Optimal policy). *Given an initial state $s_1$, the optimal policy satisfies: $\pi^*(\tau \mid s_1) \propto \mu(\tau \mid s_1)e^{\beta \sum_{t=1}^{T} Q_\zeta(s_t, a_t)}$, wherein $\tau := (s_1, a_1, \cdots, s_T, a_T)$, $\mu(\tau \mid s_1)$ represents the behavior policy and $Q_\zeta(\cdot, \cdot)$ refers to the estimated Q-value function. $\beta \geq 0$ signifies the inverse temperature controlling the energy strength.*

---

By rewriting $p_0 := \pi^*$, $q_0 = \mu$ and $\tau_0 = \tau$, we can reformulate the optimal planning into the following diffusion sampling problem:

$$p_0(\tau_0 \mid s_1) \propto q_0(\tau_0 \mid s_1) \exp\left(-\beta \mathcal{E}(\tau_0, s_1)\right), \tag{9}$$

where $\mathcal{E}(\tau_0, s_1) := -\sum_{t=1}^{T} Q_\zeta(s_t, a_t)$. Similarlly, the time-dependent energy model, $f_\eta(\tau_k, s_1, k)$, can then be trained by minizing the following contrastive loss:

$$\min_\eta \mathbb{E}_{p(k,s1)} \mathbb{E}_{\prod_{i=1}^{M} q\left(\tau_0^{(i)} \mid s_1\right) p\left(\epsilon^{(i)}\right)} \left[ -\sum_{i=1}^{M} \frac{e^{-\beta \mathcal{E}_0(h(\tau_0^{(i)}, s_1))}}{\sum_{j=1}^{M} e^{-\beta \mathcal{E}_0(h(\tau_0^{(j)}, s_1))}} \log \frac{e^{f_\eta\left(\tau_k^{(i)}, s_1, k\right)}}{\sum_{j=1}^{M} e^{f_\eta\left(\tau_k^{(j)}, s_1, k\right)}} \right], \tag{10}$$

where $k \sim \mathcal{U}(0, K)$, $\tau_k = \alpha_k \tau_0 + \sigma_k \epsilon$, and $\epsilon \sim \mathcal{N}(0, \mathbb{I})$. To estimate true distribution $q(\tau_0 \mid s_1)$ in Equation 10, we can utilize the diffusion model adopted in Diffuser or Decision Diffuser to generate $M$ support trajectories $\{\hat{\tau}_0^{(i)}\}_M$ for each initial state $s_1$ by diffusion sampling. The contrastive loss in Equation (10) is then estimated by:

$$\min_{\eta} \mathbb{E}_{k,s_1,\epsilon} - \sum_{i=1}^{M} \frac{e^{-\beta \mathcal{E}_0(h(\hat{\tau}_0^{(i)}, s_1))}}{\sum_{j=1}^{M} e^{-\beta \mathcal{E}_0(h(\hat{\tau}_0^{(j)}, s_1))}} \log \frac{e^{f_\eta \left(\hat{\tau}_k^{(i)}, s_1, k\right)}}{\sum_{j=1}^{M} e^{f_\eta \left(\hat{\tau}_k^{(j)}, s_1, k\right)}}, \tag{11}$$

where $\hat{\tau}_0^{(i)}, \hat{\tau}_0^{(j)}$ correspond to the support trajectories for each initial state $s_1$. The training procedure is shown in Algorithm 2.

---

**Algorithm 2** Efficient Planning in the Raw Action Space

---

Initialize the diffusion model $p_\theta$ and the intermediate energy model $f_\eta$
**for** each gradient step **do**                                 ▷ Training the diffusion model
    Sample $B_1$ trajectories $\tau$ from offline dataset $\mathcal{D}$
    Sample $B_1$ Gaussian noises $\epsilon$ from $\mathcal{N}(0, \mathbf{I})$ and $B_1$ time $k$ from $\mathcal{U}(0, K)$
    Perturb $\tau_0$ according to $\tau_k := \alpha_k \tau_0 + \sigma_k \epsilon$
    Update $\{\theta\}$ with the standard *score-matching loss* in Appendix B.2
**end for**
**for** each initial state $s_1$ in offline dataset $\mathcal{D}$ **do**           ▷ Generating the support trajectories
    Sample $M$ support trajectories $\{\hat{\tau}^{(i)}\}_M$ from the pretrained diffusion model $p_\theta$
**end for**
**for** each gradient step **do**                            ▷ Training the intermediate energy model
    Sample $B_2$ initial state $s_1$ from offline dataset $\mathcal{D}$
    Sample $B_2$ Gaussian noises $\epsilon$ from $\mathcal{N}(0, \mathbf{I})$ and $B_2$ time $k$ from $\mathcal{U}(0, K)$
    Retrieve support trajectories $\{\hat{\tau}_0^{(i)}\}_M$ for each $s_1$
    Perturb $\hat{\tau}_0^{(i)}$ according to $\hat{\tau}_k^{(i)} := \alpha_k \hat{\tau}_0^{(i)} + \sigma_k \epsilon$
    Update $\{\eta\}$ based on the *contrastive loss* in Equation (11)
**end for**

---

As shown in Figure 4, it becomes evident that a pronounced performance degradation manifests in the raw action space when planning compared to the latent action space, This phenomenon is even more pronounced in longer-horizon tasks, such as AntMaze.

**Connection with Diffuser**  In Diffuser (Janner et al., 2022a), $\mathcal{E}(\tau_0, s_1)$ is defined as the return of $\tau_0$. Additionally, Diffuser uses a mean-square-error (MSE) objective to train the energy model $f_\eta(\tau_t, s_1, t)$ and use its gradient for energy guidance (Lu et al., 2023b). The training objective is:

$$\min_{\eta} \mathbb{E}_{q_{0t}(\tau_0, \tau_t, s_1)} \left[ \| f_\eta \left( \tau_t, s_1, t \right) - \mathcal{E} \left( \tau_0, s_1 \right) \|_2^2 \right]. \tag{12}$$

Given the unlimited model capacity, the optimal $f_\eta$ satisfies:

$$f_\eta^{\text{MSE}} \left( \tau_t, s_1, t \right) = \mathbb{E}_{q_{0t}(\tau_0 | \tau_t, s_1)} \left[ \mathcal{E} \left( \tau_0, s_1 \right) \right]. \tag{13}$$

However, according to Lu et al. (2023, §4.1), the true energy function satifies

$$\begin{aligned} \mathcal{E}_t \left( \tau_0, s_1 \right) &= -\log \mathbb{E}_{q_{0t}(\tau_0 | \tau_t, s_1)} \left[ e^{-\mathcal{E}(\tau_0, s_1)} \right] \\ &\geq \mathbb{E}_{q_{0t}(\tau_0 | \tau_t, s_1)} \left[ \mathcal{E} \left( \tau_0, s_1 \right) \right] = f_\eta^{\text{MSE}} \left( \tau_t, s_1, t \right), \end{aligned} \tag{14}$$

and the equality only holds when $t = 0$. Therefore, the MSE energy function $f_\eta^{\text{MSE}}$ is inexact for all $t > 0$. Moreover, Lu et al. (2023) also shows that the gradient of $f_\eta^{\text{MSE}}$ is also inexact against the true gradience $\nabla_{\tau_t} \mathcal{E}_t \left( \tau_t, s_1 \right)$.

We replace the definition of $\mathcal{E}(\boldsymbol{z}_0, s_1)$ in `LatentDiffuser` with the return (or cumulative rewards) employed in Diffuser, culminating in the numerical results depicted in Figure 4. It is imperative to note that incorporating return into `LatentDiffuser` contravenes Theorem 1. As discernible in the figure, a conspicuous performance degradation ensues from the replacement. While return

bears resemblance to cumulative state-action value, the latter demonstrates diminished variance. This advantage becomes increasingly pronounced in the longer-horizon task.

In addition, we can implement an alteration to the sampling method employed by the Diffuser, thereby rendering it to an exact sampling technique. More concretely, we add an exponential activation in the original MSE-based loss in Equation (13), which is named E-MSE in (Lu et al., 2023b):

$$\min_{\eta} \mathbb{E}_{t,\boldsymbol{z}_0,\boldsymbol{z}_t} \left[ \|\exp\left(f_\eta\left(\boldsymbol{z}_t, t\right)\right) - \exp\left(\beta \mathcal{E}\left(\boldsymbol{z}_0\right)\right)\|_2^2 \right].$$

In the `LatentDiffuser`, we substitute contrastive loss with E-MSE, as depicted in Figure 4. Although E-MSE belongs to the realm of exact sampling methods, its inherent exponential terms precipitate significant numerical instability during training. Evidently, from Figure 4, the employment of E-MSE has culminated in a conspicuous decline in performance—a finding that resonates with the conclusions drawn in the Lu et al. (2023b, §H).

### F.2 PLANNING IN THE SKILL SPACE

The `LatentDiffuser` framework can be equally implemented within other variants of the latent action space. Considering the following simplified trajectory $\tau_{\text{sim}}$ of length $T$, sampled from an MDP with a fixed stochastic behavior policy, consisting of a sequence of states, and actions:

$$\tau_{\text{sim}} := (s_1, a_1, s_2, a_2, \ldots, s_T). \tag{15}$$

Under this setting, the concept of a latent action aligns perfectly with the definition of *skill* as delineated in prevailing works, although we persist in utilizing the `LatentDiffuser` framework for efficient planning within the skill space.

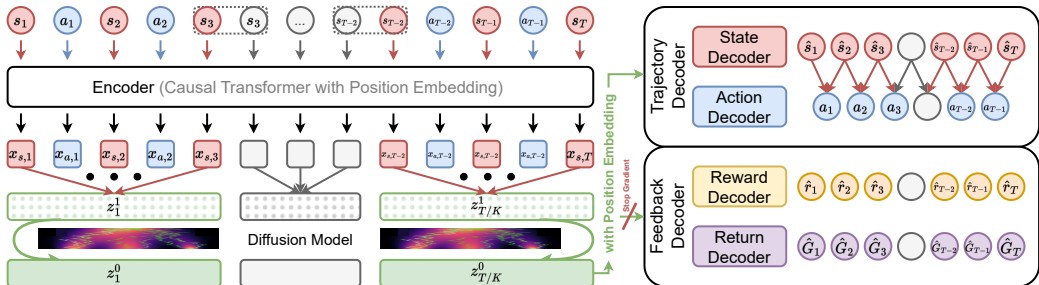

Figure 5: Skill modeling with the score-based diffusion probabilistic model.

More concretely, we can merely substitute the latent diffusion model with an variant of the latent diffusion model, as shown in Figure 5, for estimating simplified trajectory distributions. Similarily, we also can deduce a theorem, akin to Theorem 1, within the skill space:

> **Theorem 3** (Optimal skill-based policy). *Given an initial state $s_1$, the optimal skill-based policy satisfies: $\pi^*(\tau_{sim} \mid s_1) \propto \mu(\tau_{sim} \mid s_1) e^{\beta \sum_{t=1}^{T} Q_\zeta(s_t, a_t)}$, wherein $\mu(\tau_{sim} \mid s_1)$ represents the behavior policy and $Q_\zeta(\cdot, \cdot)$ refers to the estimated Q-value function. $\beta \geq 0$ signifies the inverse temperature controlling the energy strength.*

Subsequently, we can employ the algorithm nearly identical to Algorithm 1 for both the model training and the sampling of optimal trajectories.

To ascertain the efficacy of `LatentDiffuser` in skill space planning, we exchanged the latent diffusion model depicted in Figure 2 with the one shown in Figure 5. The experimental results can be observed in Figure 4. Evident from the illustration, a lack of encoding for future information (i.e., the reward and return) precipitates a significant decline in skill space planning performance as compared to that within latent action space. Similarly, this circumstance becomes increasingly pronounced in longer-horizon tasks accompanied by sparse rewards.

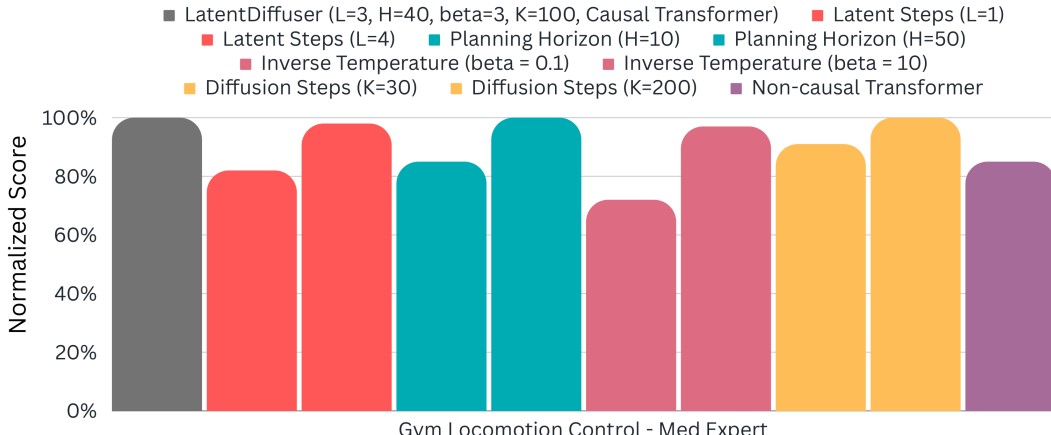

Figure 6: Results of ablation studies, where the height of the bar is the mean normalised scores on different tasks.

## G ABLATION STUDIES

In this section, ablation studies are performed on crucial hyperparameters within the `LatentDiffuser`, specifically focusing on the *latent steps* (i.e., the timesteps corresponding to the original trajectory of the latent action), the *planning horizon* (referring to the latent rather than the raw action), the *inverse temperature* (impacting energy guidance intensity), and the *diffusion steps* (contributing to the reconstructed trajectory quality). Numerical experiments were conducted (refer to Figure 6) on the *Med-Expert* dataset within the *Gym locomotion control tasks*, yielding the subsequent significant findings:

**Latent Steps** The `LatentDiffuser`'s planning occurs in a latent action space featuring temporal abstraction. When a single latent action is sampled, $L$ transitional steps extending the raw trajectory can be decoded. This design enhances planning efficiency as it reduces the number of unrolling steps to $\frac{1}{L}$; hence, the search space size is exponentially diminished. Nonetheless, the repercussions of this design on the decision-making remain uncertain. Therefore, we evaluated the `LatentDiffuser` employing varying latent steps $L$. The red bars in Figure 6 demonstrate that the reduction in latent steps $L$ to 1 leads to a substantial performance degradation. We conjecture that this performance decline is attributed to VAE overfitting, as a higher prediction error has been observed with the reduced latent step, similar with TAP (Jiang et al., 2023)

**Inverse Temperature** The energy guidance effect is regulated by the inverse temperature; decreasing values yield sampled trajectories more aligned with the behavior policy, while elevated values amplify the influence of energy guidance. As displayed by the pink bars in Figure 6, the `LatentDiffuser`'s performance noticeably deteriorates when the inverse temperature is comparably low. Alternatively, a trivial decline occurs as the value substantially increases. We propose two plausible explanations: firstly, overwhelming energy guidance may generate discrepancies between trajectory distributions, guided by energy and induced by behavior policy, negatively impacting generated quality; secondly, the energy guidance originates from an estimated intermediate energy model, which is inherently prone to overfitting throughout training, leading to inaccuracies in the estimated energy and ultimately degrading the sampling quality.

**Planning Horizon and Diffusion Steps** As demonstrated by the blue and yellow bars in Figure 6, the `LatentDiffuser` exhibits low sensitivity to variations in the planning horizon and diffusion steps. Moreover, the conclusion regarding the planning horizon may be task-specific since dense-reward locomotion control may necessitate shorter-horizon reasoning than more intricate decision-making problems. The ablations of MuZero (Hamrick et al., 2021) and TAP (Jiang et al., 2023) further reveal that real-time planning is not as beneficial in more reactive tasks, such as Atari and locomotion control.

**Transformers**   In the design of `LatentDiffuser`, we follow the settings of most existing generative methods (such as DT, TT, DD, TAP, etc.) for the encoder part, using GPT-2 style casual transformers for parameterization. Of course, in addition to this reason, another part of the reason is due to the modeling of latent actions. In the default setting, latent action consists of multiple timesteps of state, actions, rewards, and reward-to-go. We believe casual transformers will make the learned latent action representations more predictive. For example, predicting actions based on the state, predicting rewards and reward-to-go based on the state and action. This predictive ability, similar to model-based methods, will make the learned latent action representations more conducive to high-quality planning. To verify this point, we conduct comparative experiments using non-causal transformers. Specifically, we remove the mask part of the causal transformer. This means that during the encoding and decoding process, we allow the model to use the information of the entire subtrajectory to reconstruct any element within that subtrajectory, such as the state using state information. As shown in Figure 6, the experimental results show that `LatentDiffuser` has a significant performance degradation. Furthermore, we found that using a non-casual transformer is close to the performance when the latent step equals 1. These results are consistent with our previous analysis, and when using a non-casual transformer, the model is also prone to overfitting, causing the learned latent action representations to contain less information, losing a certain degree of "predictability."

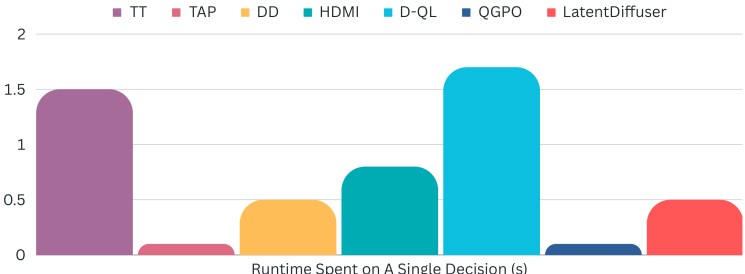

Figure 7: The average runtime spent on a single decision of baselines based on the generative model.

**Runtime**   To eliminate the influence of different algorithm implementation logic on runtime and focus on the model itself, we follow the settings of previous work and record the average time taken for different baselines to make the final action from the current input state 50 times. In the interest of fairness, we have only compared the runtime of generative methods. The final results are shown in Figure 7. As can be seen from the figure, the runtime of `LatentDiffuser` is at the average level, and the time required for making one decision is about 0.5 seconds, which is similar to DD. Although the sampling efficiency of the diffusion model has always been its weakness, we adopted the warm-up technique proposed by Diffuser, which can significantly shorten the sampling time without affecting performance. D-QL has the most extended runtime, requiring multiple samplings (50 times) to select the best result. HDMI significantly increases runtime because it is a two-layer method requiring two diffusion samplings for making one decision. TT method has a longer runtime due to its tokenized data processing, which requires longer autoregressive sequence generation before generating an action. TAP and D-QL have the shortest runtimes, with the former using beam search for planning, which can be completed quickly with a predetermined budget, but planning effectiveness is also constrained by the budget; the latter only needs to generate a one-timestep action rather than a sequence, so its runtime is also shorter. However, its final performance is significantly lower due to the lack of a planning step.

## H   LATENT ACTION VISUALIZATION

In order to gain a more intuitive understanding of the latent action space learned by `LatentDiffuser`, this section presents a visualization of the latent actions and the corresponding trajectories obtained by decoding them. Specifically, we use the fully trained `LatentDiffuser` in the `Hopper` task to sample 5 trajectories and apply the t-SNE method to reduce the dimensionality of the latent actions associated with these 5 trajectories for visualization, as shown in Figure 8. In the visualization of the trajectories, a random trajectory is selected. To facilitate presentation (due to

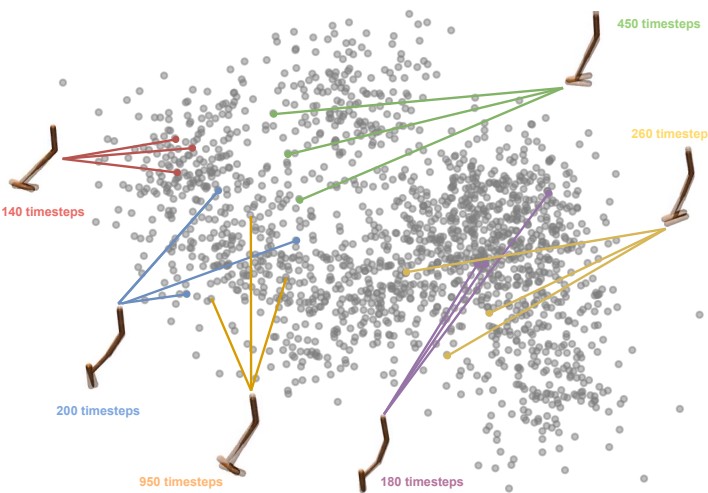

Figure 8: Visualization of latent actions and decoded trajectories.

the similarity in the robot shape at adjacent timesteps), we downsampled this trajectory by taking one latent action every 5 latent actions and overlaid the trajectory images obtained by decoding the adjacent 3 latent actions with different opacities to achieve the trajectory displayed in Figure 8. It can be seen from the figure that the latent action space learned by `LatentDiffuser` is a more compact action space, which to some extent has learned a certain type of macro-action or "skill."

# I MISSING DERIVATIONS

## I.1 PROOF OF THEOREM 1

*Proof.* Previous works (Peters et al., 2010; Peng et al., 2019) formulate offline RL as constrained policy optimization:

$$\max_{\pi} \mathbb{E}_{s \sim \mathcal{D}^{\mu}} \left[ \mathbb{E}_{a \sim \pi(\cdot|s)} A_{\zeta}(s,a) - \frac{1}{\beta} D_{\mathrm{KL}}(\pi(\cdot \mid s) \| \mu(\cdot \mid s)) \right],$$

where $A_{\zeta}$ is the action evaluation model which indicates the quality of decision $(s,a)$ by estimating the advantange function $A^{\pi}(s,a) := Q^{\pi}(s,a) - V^{\pi}(s)$ of the current policy $\pi$. $\beta$ is an inverse temperature coefficient. The first term intends to perform policy optimization, while the second term stands for policy constraint. It is shown that the optimal policy $\pi^*$ satisfies (Peng et al., 2019):

$$\pi^*(a \mid s) \propto \mu(a \mid s) e^{\beta A_{\zeta}(s,a)}.$$

Since $V^{\pi}(s)$ has nothing to do with the action $a$, the above formula can be further simplified to:

$$\pi^*(a \mid s) \propto \mu(a \mid s) e^{\beta A_{\zeta}(s,a)} \propto \mu(a \mid s) e^{\beta Q_{\zeta}(s,a)},$$

where $Q_{\zeta}$ is the action evaluation model which indicates the quality of decision $(s,a)$ by estimating the $Q$-value function $Q^{\pi}(s,a)$ of the current policy $\pi$. To simplify notation, here we reuse $\zeta$ to parameterize the estimated $Q$-value function. Furthermore, we can extend the above conclusion from raw action space, single step level to latent action space, trajectory level:

$$
\pi^*(\boldsymbol{z} \mid s_1) = p(s_1) \prod_{t=0}^{T-1} \pi^*(a_t \mid s_t) p(s_{t+1} \mid s_t, a_t) \mathcal{R}(r_t \mid s_t, a_t, s_{t+1}) \mathcal{G}(G_t \mid s_t, a_t)
$$

$$
\propto p(s_1) \prod_{t=0}^{T-1} \mu(a_t \mid s_t) \exp\left(\beta Q_\zeta(s_t, a_t)\right) p(s_{t+1} \mid s_t, a_t) \mathcal{R}(r_t \mid s_t, a_t, s_{t+1}) \mathcal{G}(G_t \mid s_t, a_t)
$$

$$
= p(s_1) \left( \prod_{t=0}^{T-1} \mu(a_t \mid s_t) p(s_{t+1} \mid s_t, a_t) \mathcal{R}(r_t \mid s_t, a_t, s_{t+1}) \mathcal{G}(G_t \mid s_t, a_t) \right) \left( \prod_{t=0}^{T-1} \exp\left(\beta Q_\zeta(s_t, a_t)\right) \right)
$$

$$
= \mu(\boldsymbol{z} \mid s_1) \exp\left( \beta \sum_{t=0}^{T-1} Q_\zeta(s_t, a_t) \right) = \mu(\boldsymbol{z} \mid s_1) e^{\beta \sum_{t=0}^{T-1} Q_\zeta(s_t, a_t)}.
$$

$\square$

### I.2   PROOF OF THEOREM 2 AND THEOREM 3

The proof procedure of Theorem 2 and Theorem 3 is similar with Theorem 1.

# REFERENCES FOR SUPPLEMENTARY MATERIAL

Suzan Ece Ada, Erhan Oztop, and Emre Ugur. Diffusion policies for out-of-distribution generalization in offline reinforcement learning. *arXiv preprint arXiv:2307.04726*, 2023.

Anurag Ajay, Yilun Du, Abhi Gupta, Joshua B Tenenbaum, Tommi S Jaakkola, and Pulkit Agrawal. Is conditional generative modeling all you need for decision-making? In *ICLR*, 2023.

Arthur Allshire, Roberto Martín-Martín, Charles Lin, Shawn Manuel, Silvio Savarese, and Animesh Garg. Laser: Learning a latent action space for efficient reinforcement learning. In *ICRA*, 2021.

Fan Bao, Min Zhao, Zhongkai Hao, Peiyao Li, Chongxuan Li, and Jun Zhu. Equivariant energy-guided SDE for inverse molecular design. In *ICLR*, 2023.

Prajjwal Bhargava, Rohan Chitnis, Alborz Geramifard, Shagun Sodhani, and Amy Zhang. Sequence modeling is a robust contender for offline reinforcement learning. *arXiv preprint arXiv:2305.14550*, 2023.

Adam Block, Daniel Pfrommer, and Max Simchowitz. Imitating complex trajectories: Bridging low-level stability and high-level behavior. *arXiv preprint arXiv:2307.14619*, 2023.

Hongyi Chen, Yilun Du, Yiye Chen, Joshua B. Tenenbaum, and Patricio A. Vela. Planning with sequence models through iterative energy minimization. In *ICLR*, 2023a.

Huayu Chen, Cheng Lu, Chengyang Ying, Hang Su, and Jun Zhu. Offline reinforcement learning via high-fidelity generative behavior modeling. In *ICLR*, 2023b.

Lili Chen, Kevin Lu, Aravind Rajeswaran, Kimin Lee, Aditya Grover, Misha Laskin, Pieter Abbeel, Aravind Srinivas, and Igor Mordatch. Decision transformer: Reinforcement learning via sequence modeling. In *NeurIPS*, 2021.

Xi Chen, Ali Ghadirzadeh, Tianhe Yu, Jianhao Wang, Alex Yuan Gao, Wenzhe Li, Liang Bin, Chelsea Finn, and Chongjie Zhang. Lapo: Latent-variable advantage-weighted policy optimization for offline reinforcement learning. In *NeurIPS*, 2022.

Rohan Chitnis, Yingchen Xu, Bobak Hashemi, Lucas Lehnert, Urun Dogan, Zheqing Zhu, and Olivier Delalleau. Iql-td-mpc: Implicit q-learning for hierarchical model predictive control. *arXiv preprint arXiv:2306.00867*, 2023.

Hyungjin Chung, Jeongsol Kim, Michael Thompson Mccann, Marc Louis Klasky, and Jong Chul Ye. Diffusion posterior sampling for general noisy inverse problems. In *ICLR*, 2023.

Robert Dadashi, Léonard Hussenot, Damien Vincent, Sertan Girgin, Anton Raichuk, Matthieu Geist, and Olivier Pietquin. Continuous control with action quantization from demonstrations. In *ICML*, 2022.

Damien Ernst, Pierre Geurts, and Louis Wehenkel. Tree-based batch mode reinforcement learning. *Journal of Machine Learning Research*, 6, 2005.

Benjamin Eysenbach, Alexander Khazatsky, Sergey Levine, and Russ R Salakhutdinov. Mismatched no more: Joint model-policy optimization for model-based rl. In *NeurIPS*, 2022.

Scott Fujimoto and Shixiang Shane Gu. A minimalist approach to offline reinforcement learning. In *NeurIPS*, 2021.

Scott Fujimoto, David Meger, and Doina Precup. Off-policy deep reinforcement learning without exploration. In *ICML*, 2019.

Wonjoon Goo and Scott Niekum. Know your boundaries: The necessity of explicit behavioral cloning in offline rl. *arXiv preprint arXiv:2206.00695*, 2022.

Alexandros Graikos, Nikolay Malkin, Nebojsa Jojic, and Dimitris Samaras. Diffusion models as plug-and-play priors. In *NeurIPS*, 2022.

Danijar Hafner, Timothy Lillicrap, Ian Fischer, Ruben Villegas, David Ha, Honglak Lee, and James Davidson. Learning latent dynamics for planning from pixels. In *ICML*, 2019.

Danijar Hafner, Timothy P Lillicrap, Mohammad Norouzi, and Jimmy Ba. Mastering atari with discrete world models. In *ICLR*, 2021.

Danijar Hafner, Jurgis Pasukonis, Jimmy Ba, and Timothy Lillicrap. Mastering diverse domains through world models. *arXiv preprint arXiv:2301.04104*, 2023.

Jessica B Hamrick, Abram L. Friesen, Feryal Behbahani, Arthur Guez, Fabio Viola, Sims Witherspoon, Thomas Anthony, Lars Holger Buesing, Petar Veličković, and Theophane Weber. On the role of planning in model-based deep reinforcement learning. In *ICLR*, 2021.

Haoran He, Chenjia Bai, Kang Xu, Zhuoran Yang, Weinan Zhang, Dong Wang, Bin Zhao, and Xuelong Li. Diffusion model is an effective planner and data synthesizer for multi-task reinforcement learning. *arXiv preprint arXiv:2305.18459*, 2023.

Dan Hendrycks and Kevin Gimpel. Gaussian error linear units (gelus). *arXiv preprint arXiv:1606.08415*, 2016.

Jonathan Ho and Tim Salimans. Classifier-free diffusion guidance. *arXiv preprint arXiv:2207.12598*, 2022.

Jifeng Hu, Yanchao Sun, Sili Huang, SiYuan Guo, Hechang Chen, Li Shen, Lichao Sun, Yi Chang, and Dacheng Tao. Instructed diffuser with temporal condition guidance for offline reinforcement learning. *arXiv preprint arXiv:2306.04875*, 2023.

Michael Janner, Justin Fu, Marvin Zhang, and Sergey Levine. When to trust your model: Model-based policy optimization. In *NeurIPS*, 2019.

Michael Janner, Qiyang Li, and Sergey Levine. Offline reinforcement learning as one big sequence modeling problem. In *NeurIPS*, 2021.

Michael Janner, Yilun Du, Joshua Tenenbaum, and Sergey Levine. Planning with diffusion for flexible behavior synthesis. In *ICML*, 2022a.

Michael Janner, Yilun Du, Joshua B Tenenbaum, and Sergey Levine. Planning with diffusion for flexible behavior synthesis. In *ICML*, 2022b.

Zhiwei Jia, Fangchen Liu, Vineet Thumuluri, Linghao Chen, Zhiao Huang, and Hao Su. Chain-of-thought predictive control. *arXiv preprint arXiv:2304.00776*, 2023.

Zhengyao Jiang, Tianjun Zhang, Michael Janner, Yueying Li, Tim Rocktäschel, Edward Grefenstette, and Yuandong Tian. Efficient planning in a compact latent action space. In *ICLR*, 2023.

Siddharth Karamcheti, Albert J Zhai, Dylan P Losey, and Dorsa Sadigh. Learning visually guided latent actions for assistive teleoperation. In *L4DC*, 2021.

Bahjat Kawar, Michael Elad, Stefano Ermon, and Jiaming Song. Denoising diffusion restoration models. In *NeurIPS*, 2022.

Rahul Kidambi, Aravind Rajeswaran, Praneeth Netrapalli, and Thorsten Joachims. Morel: Model-based offline reinforcement learning. In *NeurIPS*, 2020a.

Rahul Kidambi, Aravind Rajeswaran, Praneeth Netrapalli, and Thorsten Joachims. Morel: Model-based offline reinforcement learning. In *NeurIPS*, 2020b.

Ilya Kostrikov, Ashvin Nair, and Sergey Levine. Offline reinforcement learning with implicit q-learning. In *ICLR*, 2022.

Aviral Kumar, Aurick Zhou, George Tucker, and Sergey Levine. Conservative q-learning for offline reinforcement learning. In *NeurIPS*, 2020.

Sergey Levine. Reinforcement learning and control as probabilistic inference: Tutorial and review. *arXiv preprint arXiv:1805.00909*, 2018.

Sergey Levine, Aviral Kumar, George Tucker, and Justin Fu. Offline reinforcement learning: Tutorial, review, and perspectives on open problems. *arXiv preprint arXiv:2005.01643*, 2020.

Wenhao Li, Xiangfeng Wang, Bo Jin, and Hongyuan Zha. Hierarchical diffusion for offline decision making. In *ICML*, 2023.

Zhixuan Liang, Yao Mu, Mingyu Ding, Fei Ni, Masayoshi Tomizuka, and Ping Luo. Adaptdiffuser: Diffusion models as adaptive self-evolving planners. In *ICML*, 2023.

Dylan P Losey, Hong Jun Jeon, Mengxi Li, Krishnan Srinivasan, Ajay Mandlekar, Animesh Garg, Jeannette Bohg, and Dorsa Sadigh. Learning latent actions to control assistive robots. *Autonomous robots*, 46(1):115–147, 2022.

Cheng Lu, Yuhao Zhou, Fan Bao, Jianfei Chen, Chongxuan Li, and Jun Zhu. Dpm-solver: A fast ode solver for diffusion probabilistic model sampling in around 10 steps. In *NeurIPS*, 2022a.

Cheng Lu, Yuhao Zhou, Fan Bao, Jianfei Chen, Chongxuan Li, and Jun Zhu. Dpm-solver++: Fast solver for guided sampling of diffusion probabilistic models. *arXiv preprint arXiv:2211.01095*, 2022b.

Cheng Lu, Huayu Chen, Jianfei Chen, Hang Su, Chongxuan Li, and Jun Zhu. Contrastive energy prediction for exact energy-guided diffusion sampling in offline reinforcement learning. In *ICML*, 2023a.

Cheng Lu, Huayu Chen, Jianfei Chen, Hang Su, Chongxuan Li, and Jun Zhu. Contrastive energy prediction for exact energy-guided diffusion sampling in offline reinforcement learning. In *ICML*, 2023b.

Cong Lu, Philip Ball, Jack Parker-Holder, Michael Osborne, and Stephen J Roberts. Revisiting design choices in offline model based reinforcement learning. In *ICLR*, 2022c.

Kevin Lu, Aditya Grover, Pieter Abbeel, and Igor Mordatch. Reset-free lifelong learning with skill-space planning. In *ICLR*, 2021.

J Merel, L Hasenclever, A Galashov, A Ahuja, V Pham, G Wayne, Y Teh, and N Heess. Neural probabilistic motor primitives for humanoid control. In *International Conference on Learning Representations*, 2019.

Diganta Misra. Mish: A self regularized non-monotonic neural activation function. *arXiv preprint arXiv:1908.08681*, 2019.

Ashvin Nair, Abhishek Gupta, Murtaza Dalal, and Sergey Levine. Awac: Accelerating online reinforcement learning with offline datasets. *arXiv preprint arXiv:2006.09359*, 2020.

Fei Ni, Jianye Hao, Yao Mu, Yifu Yuan, Yan Zheng, Bin Wang, and Zhixuan Liang. Metadiffuser: Diffusion model as conditional planner for offline meta-rl. In *ICML*, 2023.

Alexander Quinn Nichol and Prafulla Dhariwal. Improved denoising diffusion probabilistic models. In *ICML*, 2021.

Sherjil Ozair, Yazhe Li, Ali Razavi, Ioannis Antonoglou, Aaron Van Den Oord, and Oriol Vinyals. Vector quantized models for planning. In *ICML*, 2021.

Tim Pearce, Tabish Rashid, Anssi Kanervisto, Dave Bignell, Mingfei Sun, Raluca Georgescu, Sergio Valcarcel Macua, Shan Zheng Tan, Ida Momennejad, Katja Hofmann, and Sam Devlin. Imitating human behaviour with diffusion models. In *ICLR*, 2023.

Xue Bin Peng, Aviral Kumar, Grace Zhang, and Sergey Levine. Advantage-weighted regression: Simple and scalable off-policy reinforcement learning. *arXiv preprint arXiv:1910.00177*, 2019.

Xue Bin Peng, Yunrong Guo, Lina Halper, Sergey Levine, and Sanja Fidler. Ase: Large-scale reusable adversarial skill embeddings for physically simulated characters. *ACM Transactions On Graphics (TOG)*, 41(4):1–17, 2022.

Jan Peters, Katharina Mulling, and Yasemin Altun. Relative entropy policy search. In *AAAI*, 2010.

Ben Poole, Ajay Jain, Jonathan T. Barron, and Ben Mildenhall. Dreamfusion: Text-to-3d using 2d diffusion. In *ICLR*, 2023.

Alec Radford, Jeff Wu, Rewon Child, David Luan, Dario Amodei, and Ilya Sutskever. Language models are unsupervised multitask learners, 2019.

Juergen Schmidhuber. Reinforcement learning upside down: Don't predict rewards–just map them to actions. *arXiv preprint arXiv:1912.02875*, 2019.

Julian Schrittwieser, Ioannis Antonoglou, Thomas Hubert, Karen Simonyan, Laurent Sifre, Simon Schmitt, Arthur Guez, Edward Lockhart, Demis Hassabis, Thore Graepel, et al. Mastering atari, go, chess and shogi by planning with a learned model. *Nature*, 588(7839):604–609, 2020.

Yang Song, Prafulla Dhariwal, Mark Chen, and Ilya Sutskever. Consistency models. In *ICML*, 2023.

HJ Suh, Glen Chou, Hongkai Dai, Lujie Yang, Abhishek Gupta, and Russ Tedrake. Fighting uncertainty with gradients: Offline reinforcement learning via diffusion score matching. *arXiv preprint arXiv:2306.14079*, 2023.

Richard S Sutton. Integrated architectures for learning, planning, and reacting based on approximating dynamic programming. In *ICML*, 1990.

Ashish Vaswani, Noam Shazeer, Niki Parmar, Jakob Uszkoreit, Llion Jones, Aidan N Gomez, Łukasz Kaiser, and Illia Polosukhin. Attention is all you need. In *NeurIPS*, 2017.

Siddarth Venkatraman. Latent skill models for offline reinforcement learning. Master's thesis, Carnegie Mellon University Pittsburgh, PA, 2023.

Jianhao Wang, Wenzhe Li, Haozhe Jiang, Guangxiang Zhu, Siyuan Li, and Chongjie Zhang. Offline reinforcement learning with reverse model-based imagination. In *NeurIPS*, 2021.

Linnan Wang, Rodrigo Fonseca, and Yuandong Tian. Learning search space partition for black-box optimization using monte carlo tree search. In *NeurIPS*, 2020a.

Zhendong Wang, Jonathan J Hunt, and Mingyuan Zhou. Diffusion policies as an expressive policy class for offline reinforcement learning. In *ICLR*, 2023.

Ziyu Wang, Alexander Novikov, Konrad Zolna, Josh S Merel, Jost Tobias Springenberg, Scott E Reed, Bobak Shahriari, Noah Siegel, Caglar Gulcehre, Nicolas Heess, et al. Critic regularized regression. In *NeurIPS*, 2020b.

Yueh-Hua Wu, Xiaolong Wang, and Masashi Hamaya. Elastic decision transformer. *arXiv preprint arXiv:2307.02484*, 2023.

Yuxin Wu and Kaiming He. Group normalization. In *ECCV*, 2018.

Kevin Yang, Tianjun Zhang, Chris Cummins, Brandon Cui, Benoit Steiner, Linnan Wang, Joseph E Gonzalez, Dan Klein, and Yuandong Tian. Learning space partitions for path planning. In *NeurIPS*, 2021.

Min Zhao, Fan Bao, Chongxuan Li, and Jun Zhu. Egsde: Unpaired image-to-image translation via energy-guided stochastic differential equations. In *NeurIPS*, 2022.

Qinqing Zheng, Amy Zhang, and Aditya Grover. Online decision transformer. In *ICML*, 2022.

Wenxuan Zhou, Sujay Bajracharya, and David Held. Plas: Latent action space for offline reinforcement learning. In *CoRL*, 2021.

Zhaoyi Zhou, Chuning Zhu, Runlong Zhou, Qiwen Cui, Abhishek Gupta, and Simon Shaolei Du. Free from bellman completeness: Trajectory stitching via model-based return-conditioned supervised learning. *arXiv preprint arXiv:2310.19308*, 2023.

