# OpenReview forum: "Efficient Planning with Latent Diffusion"
_ICLR.cc/2024/Conference — ICLR 2024 poster_

### Official Review · Reviewer_z1ps · 2023-10-30

**Soundness:** 4 excellent
**Presentation:** 4 excellent
**Contribution:** 4 excellent
**Rating:** 8
**Confidence:** 5

**Summary:**

The authors propose an algorithm to leverage latent diffusion for planning skills in offline RL. Similar to previous works using latent diffusion, this work first trains a transformer encoder/decoder with weak KL regularization to model trajectory sequences, and then trains a diffusion prior to sample these latents. Then, an energy guided sampling method is proposed to plan in this latent skill space through guided diffusion sampling. The approach is evaluated on the D4RL benchmark, where it achieves strong results.

**Strengths:**

1) Leveraging latent diffusion for offline RL is a very natural extension to previous raw trajectory diffusion works such as Diffuser and Decision Diffuser. Diffusion planning latent space leverages the hierarchical structure of the planning problem very effectively, leaving the low level action prediction to the skill conditioned decoder instead.

2) The experimental results on D4RL are strong and convincing.

3) The work is highly relevant to the growing interest in viewing reinforcement learning through the lens of conditional generative modeling.

4) The paper is written very clearly, with good motivation presented.

**Weaknesses:**

While I acknowledged the potential strength of hierarchical planning above, since the latent space is weakly regularized, we should not expect there to be too much benefit in leveraging this space for planning as compared to the raw trajectory space. Unlike the case of latent diffusion for image generative modeling where we simply require high quality samples with improved efficiency, here the primary motivation for leveraging latent planning is to plan with higher level skills.

**Questions:**

1) Could the authors clarify why they expect a weakly regularized latent space to be useful for planning as compared to the raw actions? I see in the appendix that the comparison is made, and raw actions indeed are shown to perform worse. But I would like some intuition for why this should be the case, and if some better hyperparameter tuning could have produced similar results with raw actions.

2) The authors use the return to go predictions from the decoder as Q values. Since TD learning is not done to learn optimal Q functions, these Q values correspond to those of the behavior policy. Is there a reason some form of offline Q learning technique such as CQL/BCQ/IQL was not done? Without doing Q learning, should we expect skill stitching to occur?

I would also encourage the authors to cite a recent paper from Venkatraman et al. [1], which proposes a very similar algorithm. I am happy to recommend this paper for acceptance, primarily due to its strong empirical results and relevance to the current trends in offline RL.

[1] Reasoning with Latent Diffusion in Offline Reinforcement Learning, arxiv.org/abs/2309.06599

---

> ### Author Response · Authors · 2023-11-15
>
> We greatly appreciate your taking the time to carefully review our paper and provide valuable suggestions and constructive feedback. This has significantly helped us improve the rigor and completeness of our paper. Additionally, we are grateful for your recognition of our work! Below, we have organized your questions and will address them individually.
>
> `Q1: Could the authors clarify why they expect a weakly regularized latent space to be helpful in planning compared to the raw actions? I see in the appendix that the comparison is made, and raw actions are shown to perform worse. But I would like some intuition for why this should be the case and if some better hyperparameter tuning could have produced similar results with raw actions.`
>
> Thank you very much for your question! This is a great question, and the reason behind it is not intuitive. In the default setting, latent action consists of multiple timesteps of state, actions, rewards, and reward-to-go. Using a casual transformer as the encoder and decoder will make the learned latent action representations more predictive. For example, predicting actions based on the state, predicting rewards and reward-to-go based on the state and action. This predictive ability, similar to model-based methods, will make the learned latent action representations more conducive to high-quality planning. At the same time, a weakly regularized latent space does not have much of a negative impact on the quality of the trajectories obtained through planning. This is because the quality of the trajectories is mainly determined by diffusion sampling. LatentDiffuser uses an exact energy-guided diffusion sampling method to sample trajectories. As long as the energy is specified as the expected return value corresponding to high-quality trajectories during sampling, the quality of the sampled trajectories can be guaranteed to some extent.
>
> To verify this point, we also conducted comparative experiments using non-causal transformers. Specifically, we removed the mask part of the causal transformer. This means that during the encoding and decoding process, we allow the model to use the information of the entire subtrajectory to reconstruct any element within that subtrajectory, such as the state using state information. As shown in Figure 6 in the revised version, the experimental results show that LatentDiffuser has a significant performance degradation. Furthermore, we found that using a non-casual transformer is close to the performance when the latent step equals 1. These results are consistent with our previous analysis, and when using a non-casual transformer, the model is also prone to overfitting, causing the learned latent action representations to contain less information, losing a certain degree of "predictability."
>
> We have added the above analysis to Appendix G in the revised version. We hope this analysis can address your concerns.
>
> `Q2: The authors use the return-to-go predictions from the decoder as Q values. Since TD learning is not done to learn optimal Q functions, these Q values correspond to the behavior policies. Is there a reason some form of offline Q learning technique such as CQL/BCQ/IQL was not done? Without doing Q learning, should we expect skill stitching to occur?`
>
> Thank you very much for your question! This is a very good question!
>
> We primarily considered the following two points as reasons for not using CQL/BCQ/IQL and other TD learning-based offline RL methods:
>
> First, even without using TD learning methods, generative approaches based on diffusion models can implement implicit dynamic programming, thereby achieving trajectory stitching. This point has been preliminarily verified through a simple experiment in Appendix A.1 of the Decision Diffuser paper [1]. We also conducted the same experiment in Appendix D.1 of the revised version and obtained similar results. Of course, exploring the trajectory stitching capabilities of generative approaches based on diffusion models is an intriguing and worthwhile problem for deeper investigation.
>
> Secondly, we drew upon the CEP method [2] in the exact energy-guided diffusion sampling section of LatentDiffuser. The authors proposed an offline RL method based on vanilla TD-learning called QGPO in the CEP paper. QGPO requires additional data sampled from the diffusion model during training to train the Q-function, significantly increasing training overhead. Thanks to the introduction of the latent diffusion model in LatentDiffuser, we can precisely use the pretrained decoder as an estimate for the reward-to-go, thereby improving training efficiency.
>
> We hope that the above explanation answers your question. Thanks again for your inquiry!
>
> [1] Ajay, Anurag, et al. "Is Conditional Generative Modeling All You Need for Decision Making?." *ICLR* 2022.
>
> [2] Lu, Cheng, et al. "Contrastive Energy Prediction for Exact Energy-Guided Diffusion Sampling in Offline Reinforcement Learning." *ICML* 2023.

---

> > ### Comment · Reviewer_z1ps · 2023-11-18
> > **Response to Author Comment**
> >
> > Thank you for your detailed response to my questions. I will maintain my score recommending acceptance.

---

> > > ### Author Response · Authors · 2023-11-18
> > >
> > > Once again, thank you for your kind recognition of our work! We truly appreciate your support.

---

> ### Author Response · Authors · 2023-11-15
>
> `Q3: I would also encourage the authors to cite a recent paper from Venkatraman et al. [1], which proposes a similar algorithm. I am happy to recommend this paper for acceptance, primarily due to its solid empirical results and relevance to the current trends in offline RL.`
>
> Thank you very much for your suggestions. We apologize for our oversight in this matter. Since this paper was made public on arXiv during the final stages of our work, just before the ICLR submission deadline, we have not discussed it in the related work section. At the end of Appendix C of the revised version, we set aside a separate remark section to emphasize the similarities and differences between this work and LatentDiffuser. The specific content is pasted below.
>
> Recently, Venkatraman et al. [1] proposed a novel algorithm called LDCQ, which is very similar to LatentDiffuser. Specifically, LDCQ also introduces a latent diffusion model to learn a latent action space. Unlike the latent action used by LatentDiffuser, the learned latent action space in LDCQ belongs to the skill space, similar to what is described in Appendix A. Additionally, LDCQ does not perform planning within the learned skill space but instead uses model-free TD-learning methods to choose the optimal skill at each timestep and obtains the final action using a decoder. In summary, it is quite a coincidence that LDCQ and LatentDiffuser belong to two orthogonal approaches to utilizing latent diffusion models in offline RL. The former still adopts the RL framework to model the offline RL problems, while LatentDiffuser approaches the problem from a conditional generative perspective.
>
> Lastly, we thank you once again for your recognition of our work!
>
> [1] Venkatraman, Siddarth, et al. "Reasoning with latent diffusion in offline reinforcement learning." *arXiv preprint arXiv:2309.06599* (2023).

---

### Official Review · Reviewer_A5bD · 2023-10-30

**Soundness:** 3 good
**Presentation:** 2 fair
**Contribution:** 3 good
**Rating:** 5
**Confidence:** 2

**Summary:**

This paper proposes the use of latent diffusion models for planning, especially in tasks with sparse rewards and long horizon. The key proposal is to use continuous latent action space representations for planning, through the use of latent score based diffusion models. The novel framework takes conditional generative models for planning in the latent domain, where the authors learn a latent action representation and use the latents for planning. The authors propose a multi-stage framework to address the issues of current hierarchical offline reinforcement learning algorithms. The framework utilizes diffusion models to learn latent actions from offline datasets. This paper argues the equivalence between planning with latent action representation and energy-guided sampling.

**Strengths:**

This paper tackles offline RL tasks that require temporal abstraction, and proposes the use of latent action representations and latent actions for planning based on a diffusion based model. The task is formulated as a conditional diffusion problem, conditioning on returns. The paper seems to be well derived from an algorithmic viewpoint, and is one of the early works that seem to do planning based on a latent diffusion based approach. However, in its current form the paper is quite difficult to understand (see my points on weaknesses)

The proposed method scores better or on par with the baselines on multiple benchmarks with higher dimension action space and a long horizon. The paper presents the effectiveness of adding reward-to-go values in the formulation of a trajectory to learn sequence modeling.

**Weaknesses:**

The paper repeatedly uses terms like the latent action representation and latent action planning, without a carefully derived definition of it. For self consistency, it would be helpful to define these terms more concretely; otherwise, in its current format, the contributions of the paper can be hard to follow.

I can see that the paper draws inspiration from the TAP paper (Jiang et al., 2023) and integrates a diffuser based latent step within this framework. The paper in its current form can be quite confusing to follow and difficult to understand.

The model seems overly complicated to understand and it is not clear what the contributions of the paper really are. I understand the relations with the TAP paper and how the work is trying to build from it, but I believe the planning approach based on diffusion doesn’t really have any novel approach? Am I right?

Overall, equation 4 is the objective that needs to be optimzied, with the first term that generates the planning trajectory based on the latents from the diffuser? Am I correct?

Experimental results are compared with few baselines using the D4RL setup. I believe all the results from 5.1 - 5.3 are using the same setup, although studying different aspects of the proposed algorithm? It probably would have been useful if there were much simpler proof of concept experiments, especially focusing on the planning part and how the diffuser plays a role there?

The experimental results need an explanation of performance in comparison to the baselines.

The paper requires more proofreading as it contains typos (such as “emploies”).

**Questions:**

Why does the proposed method work on par with the TAP baseline in the Expert dataset of the Adroit task?

Why was the GPT-2 style Transformer chosen over other choices of transformers as the encoder?

What is the total runtime of the proposed method in comparison to the baselines?

---

> ### Author Response · Authors · 2023-11-15
>
> We are extremely grateful for your taking the time to thoroughly review our paper and provide numerous constructive comments and suggestions. This has been of great help in enhancing the rigor and completeness of our paper. Below, we have compiled and addressed each of the questions you raised. For the sake of logical flow, we have adjusted the order of a few issues, and we hope for your understanding.
>
> `Q1: The paper repeatedly uses terms like latent action representation and latent action planning without a carefully derived definition.`
>
> We sincerely apologize for any confusion caused by our unclear description, and we greatly appreciate your suggestions! The term "latent action representation" refers to the learning of representations for latent actions. In our paper, latent actions are considered a latent variable that needs to be generated by encoding via an encoder. On the other hand, "latent action planning" refers to planning within the latent action space rather than the original space. We hope the above explanation can clear up any confusion. In the revised version, we modified the descriptions of latent action representation and latent action planning according to the above explanations.
>
> `Q2: I can see that the paper draws inspiration from the TAP paper (Jiang et al., 2023) and integrates a diffuser-based latent step within this framework. The paper in its current form can be confusing and difficult to follow. The model seems overly complicated to understand, and it is not clear what the contributions of the paper are. I understand the relations with the TAP paper and how the work is trying to build from it, but I believe the planning approach based on diffusion doesn’t have any novel approach. Am I right?`
>
> We apologize for the confusion caused due to our content organization. We hope that our response to your concern will help alleviate the problem. As you mentioned, planning based on the diffusion model is not a novel methodology. However, our paper's main contribution is not solely based on using the diffusion model for implicit planning. Specifically, we aim to introduce the concept of latent actions proposed by TAP and perform efficient planning based on the diffusion model in the latent action space. This approach differs from existing methods of implicit planning based on diffusion models.
>
> Our framework also has two advantages over TAP. First, our framework allows algorithms to plan in a continuous latent action space, enhancing the capability of addressing more complex problems. Second, unlike TAP, which separates the learning phase of latent action representation and the planning phase, our framework integrally combines these two phases to enable end-to-end learning.
>
> We introduce the recently proposed exact energy-guided diffusion sampling method CEP to achieve efficient planning in the latent action space. However, CEP uses diffusion models to parameterize a policy rather than directly generating entire trajectories, so no planning step is involved. Therefore, we establish the theoretical equivalence between planning and exact energy-guided diffusion sampling, deeply integrating CEP into the planning process based on the latent action space. We also improved the CEP algorithm. Specifically, CEP requires additional TD-learning for a Q-function to sample high-quality trajectories. In LatentDiffuser, due to the definition of latent actions, we can directly use the reward-to-go estimates output by the latent diffusion model to replace the Q-function in CEP, thereby reducing the training cost.
>
> In addition, due to the flexibility of the 'latent action' definition, our framework can also efficiently plan in the raw action and skill spaces (see Appendix F).
>
> In summary, LatentDiffuser is not a simple rearrangement of existing technologies. Although LatentDiffuser is based on existing technologies, neither the latent diffusion model nor the exact energy-guided sampling method (CEP) has been fully applied to the planning paradigm in offline RL. Each tool has its necessity and rationality for implementing efficient planning, and we have also adapted and improved these tools in a task-oriented manner. Our main contribution is precisely manifested in these tools' organic integration, deep adaptation, and improvement.

---

> ### Author Response · Authors · 2023-11-15
>
> `Q3: Overall, equation 4 is the objective that needs to be optimized, with the first term that generates the planning trajectory based on the latent from the diffuser. Am I correct?`
>
> Sure, your understanding is entirely correct. I would like to explain Equation 4 further, hoping to provide you with a more intuitive understanding of our paper on a more specific level.
>
> The reconstruction and entropy terms for the above objective function are easily estimated for any explicit encoder. The challenging aspect pertains to training the cross entropy term, which involves the score-based prior. We adopt a more straightforward yet efficacious approach by training a VAE $\{q_{\boldsymbol{\phi}},p_{\boldsymbol{\psi}}\}$ and a score-based diffusion model $\{q_{\boldsymbol{\theta}}\}$ consecutively based on the offline dataset $\mathcal{D}_{\tau}$.
>
> `Q4: Experimental results are compared with a few baselines using the D4RL setup. I believe all the results from 5.1 - 5.3 use the same setup, although studying different aspects of the proposed algorithm?`
>
> Thank you very much for your question. Yes, the hyperparameters involved in the model and training process are essentially the same for the three sets of different tasks in Sections 5.1-5.3, but there are differences for different tasks regarding the diffusion model. Specifically, for gym locomotion control and Adroit tasks, the dimension of the latent actions set, $M$, is $16$, whereas, for AntMaze tasks, it is fixed at $32$. In all tasks, however, $\beta$ is set to $3$. The planning horizon, $H$, is set to $40$ for gym locomotion control and Adroit tasks and $100$ for AntMaze tasks. The guidance scale, $w$, is selected from the range $\{1.2,1.4,1.6,1.8\}$, although the specific value depends on the task. For further details, please refer to Appendix E.2.2.
>
> `Q5: The experimental results need an explanation of performance compared to the baselines.`
>
> Thank you very much for your suggestion, which helps enhance the manuscript’s completeness. We have added the following analysis of the experimental results in Appendix D:
>
> This section will then delve into a more detailed analysis of the performance differences among different baselines across various tasks. To provide an intuitive comparison of different algorithms, we classify them from three perspectives — planning, hierarchy, and generative — according to the classification method shown in Table 5. Firstly, the Gym locomotion task has a long horizon, dense rewards, and low action dimensions, making it a baseline test task for offline RL. The results from Table 4 show that generative methods based on diffusion models generally perform better. The community currently attributes this to diffusion models' more powerful representation capabilities in modeling more complex policies or environmental models. However, LatentDiffuser does not demonstrate its advantages well in the low-dimensional action space. Although LatentDiffuser approaches the SOTA performance on this task, it is mainly due to a better diffusion sampling method, which is supported by the solid performance of the QGPO method. Due to dense rewards, planning and hierarchy-based methods, such as TAP and HDMI, have not achieved the best results.
>
> Secondly, the Adroit task is characterized by a high-dimensional action space. This leads to the best performance for TAP and LatentDiffuser (see Table 2), two methods based on latent action, which experimentally verify the effectiveness of latent action. Additionally, generative methods based on diffusion models generally exhibit better performance. However, due to the shorter horizon of the Adroit task, the HDMI method, which is based on planning and hierarchy, does not achieve the best performance.
>
> Lastly, the AntMaze task has a longer horizon and very sparse rewards. This allows latent action ample room for improvement (see Table 2). Moreover, methods based on planning and hierarchy also achieve good results, such as HDMI. In this task, non-generative methods based on planning and hierarchy, such as ComPILE and GoFAR, approach the performance of generative methods without planning and hierarchy (D-QL).
>
> We hope that the above content allows you to gain a deeper understanding of the experimental results.
>
> `Q6: The paper requires more proofreading as it contains typos (such as “emploies”).`
>
> Thank you for your thorough review! We have polished the entire text and corrected the typos in the revised version.

---

> ### Author Response · Authors · 2023-11-15
>
> `Q7: Why does the proposed method work on par with the TAP baseline in the Expert dataset of the Adroit task?`
>
> Thank you very much for your question! This is a great question, and the reason behind it is not intuitive. TAP and LatentDiffuser's performance gap on the expert dataset is smaller than on other Adroit and Gym locomotion tasks datasets. We analyzed that the primary source of this performance gap comes from the proportion of suboptimal trajectories in the dataset. In non-expert datasets, the proportion of suboptimal trajectories is more significant. To learn the optimal policy from the dataset, the algorithm needs to have the "trajectory stitch" ability, i.e., to splice segments of suboptimal trajectories to form an optimal trajectory.
>
> On the one hand, most of the current offline RL methods are based on a dynamic programming framework to learn a Q function. However, these methods require the Q function to have Bellman completeness to achieve good performance. Designing a function class with Bellman completeness is very challenging [1]. On the other hand, DD[2, Appendix A.1] has found that generative methods based on diffusion models possess implicit dynamic programming capabilities. These methods use the powerful representation ability of diffusion models to bypass Bellman completeness and achieve the "trajectory stitch" ability. This allows them to perform well in datasets with more suboptimal trajectories.
>
> LatentDiffuser is a generative method based on a diffusion model, while TAP is not. This leads to a more significant performance gap between the two on non-expert datasets. In expert datasets, however, LatentDiffuser's advantage cannot be demonstrated. We hope the above analysis can answer your doubts. We have also added this analysis to the revised version (Appendix D). Thanks again for your question!
>
> [1] Zhou, Zhaoyi, et al. "Free from Bellman Completeness: Trajectory Stitching via Model-based Return-conditioned Supervised Learning." *arXiv preprint arXiv:2310.19308* (2023).
>
> [2] Ajay, Anurag, et al. "Is Conditional Generative Modeling All You Need for Decision Making?." *ICLR* 2022.
>
> `Q8: It probably would have been useful if there were much simpler proof of concept experiments, especially focusing on the planning part and how the diffuser plays a role there.`
>
> Thank you very much for your suggestion. Since existing generative methods for solving offline RL problems primarily use Gym locomotion as a proof-of-concept benchmark, our original paper version also followed the tradition. It is somewhat challenging to develop a suitable proof-of-concept task to validate the implicit planning capabilities of the diffusion model in a short period, as planning is a very broad concept. To address this, we plan to construct a simple experiment from the perspective of trajectory stitching.
>
> Concretely, we designed an experiment similar to the Decision Diffuser to demonstrate the importance of planning. Most tasks' offline datasets contain a large number of suboptimal trajectories. To learn better policies rather than just simple behavior cloning, trajectory stitching is one of the essential abilities algorithms must possess. To validate whether the LatentDiffuser can achieve trajectory stitching through implicit planning, we adopted the same experimental setup as the Decision Diffuser.
>
> In the maze-2D-open environment, the objective is to navigate towards the target area on the right side, with the reward being the negative distance to this target area. The training dataset comprises $500$ trajectories originating from the left side and terminating at the bottom and $500$ trajectories starting from the bottom and ending at the right side. Each trajectory is constrained to a maximum length of $50$. At test time, the agent begins on the left side and aims to reach the right side as efficiently as possible. As demonstrated in Figure 3 in the revised version and consistent with the findings of the Decision Diffuser, the LatentDiffuser can effectively stitch trajectories from the training dataset to produce trajectories that traverse from the left side to the right side in (near) straight lines.
>
> We hope that this simple experiment on trajectory stitching can help you better understand the implicit planning capabilities of diffusion models and the importance of planning. We have added the aforementioned experiment to Appendix D.1.

---

> ### Author Response · Authors · 2023-11-15
>
> `Q9: Why was the GPT-2 style Transformer chosen over other choices of transformers as the encoder?`
>
> Thank you very much for your question! In the design of LatentDiffuser, we follow the settings of most existing generative methods (such as DT, TT, DD, TAP, etc.) for the encoder part, using GPT-2 style casual transformers for parameterization. Of course, in addition to this reason, another part of the reason is due to the modeling of latent actions. In the default setting, latent action consists of multiple timesteps of state, actions, rewards, and reward-to-go. We believe casual transformers will make the learned latent action representations more predictive. For example, predicting actions based on the state, predicting rewards and reward-to-go based on the state and action. This predictive ability, similar to model-based methods, will make the learned latent action representations more conducive to high-quality planning.
>
> To verify this point, we also conducted comparative experiments using non-causal transformers. Specifically, we removed the mask part of the causal transformer. This means that during the encoding and decoding process, we allow the model to use the information of the entire subtrajectory to reconstruct any element within that subtrajectory, such as the state using state information. As shown in Figure 6 in the revised version, the experimental results show that LatentDiffuser has a significant performance degradation. Furthermore, we found that using a non-casual transformer is close to the performance when the latent step equals 1. These results are consistent with our previous analysis, and when using a non-casual transformer, the model is also prone to overfitting, causing the learned latent action representations to contain less information, losing a certain degree of "predictability."
>
> We have added the above analysis to Appendix G in the revised version. We hope this analysis can address your concerns.
>
> `Q10: What is the proposed method’s total runtime compared to the baselines?`
>
> Thank you very much for your question, which helps enhance the manuscript’s completeness.To eliminate the influence of different algorithm implementation logic on runtime and focus on the model itself, we follow the settings of previous work and record the average time taken for different baselines to make the final action from the current input state 50 times. In the interest of fairness, we have only compared the runtime of generative methods. The final results are shown in Figure 7 in the Appendix of the revised version. As can be seen from the figure, the runtime of LatentDiffuser is at the average level, and the time required for making one decision is about 0.5 seconds, which is similar to DD. Although the sampling efficiency of the diffusion model has always been its weakness, we adopted the warm-up technique proposed by Diffuser, which can significantly shorten the sampling time without affecting performance. D-QL has the most extended runtime, requiring multiple samplings (50 times) to select the best result. HDMI significantly increases runtime because it is a two-layer method requiring two diffusion samplings for making one decision. TT method has a longer runtime due to its tokenized data processing, which requires longer autoregressive sequence generation before generating an action. TAP and D-QL have the shortest runtimes, with the former using beam search for planning, which can be completed quickly with a predetermined budget, but planning effectiveness is also constrained by the budget; the latter only needs to generate a one-timestep action rather than a sequence, so its runtime is also shorter. However, its final performance is significantly lower due to the lack of a planning step.
>
> We have also added the above analysis to Appendix G in the revised version. We hope this analysis can address your concerns.

---

> ### Author Response · Authors · 2023-11-21
> **Have we addressed your concerns?**
>
> Thanks again for your time and effort in reviewing our paper! As the discussion period is coming to a close, we would like to know if we have resolved your concerns expressed in the original reviews. We remain open to any further feedback and are committed to making additional improvements if needed. If you find that these concerns have been resolved, we would be grateful if you would consider reflecting this in your rating of our paper : )

---

### Official Review · Reviewer_qvER · 2023-10-31

**Soundness:** 3 good
**Presentation:** 4 excellent
**Contribution:** 3 good
**Rating:** 8
**Confidence:** 4

**Summary:**

The authors present a method that addresses offline RL using diffusion modelling. The authors combine several elements from recent work, namely using a latent representation of the path, while sampling latent variables using a diffusion model. The authors moreover make use of recent work that demonstrates effective sampling from energy based distributions. The authors demonstrate the utility of their approach on standard RL baselines and compare against a collection of other methods. They demonstrate improvement in particular for high dimensional tasks.

**Strengths:**

- The authors demonstrate improvement in the area where they expect their model to do well
 - The paper is well written
 - Elements seem to fit well together

**Weaknesses:**

- While I think this is not a very big deal, the paper relies mostly on existing methodologies

**Questions:**

Overall I enjoyed seeing the elements from previous work come together in this work. I think overall the authors did quite a good job writing this paper, although the overall notation, in particular where x and y get used, together with tau, and state tuples, can be a bit confusing. While the paper is a combination of previous work, in my opinion the algorithm the authors created is novel enough in itself to warrant publication. The authors demonstrate performance in their experiments and, importantly, provide a useful interpretation with their experiments to demonstrate specifically where they expect to make a difference. The current review system precludes 7 as a score, and until I read the rebuttal I am rounding down.
Specific points:
 - Can the authors comment on how the sequence length is determined? Is the sequence terminated externally?
Small stuff
 - A reference for the proof for theorem 1 is missing.
 - Is the text inconsistent with Fig2? It seems like the initial state in Fig 2 is labelled s0, while in the text there is only ever reference to s1
 - On page 5 typo "emploies".

---

> ### Author Response · Authors · 2023-11-15
>
> We are incredibly grateful for your willingness to take the time to conduct a thorough review of our paper and provide numerous constructive comments and suggestions. This has been immensely helpful in enhancing the rigor and completeness of our paper. Additionally, we greatly appreciate your recognition of our work! Below, we have organized the questions you raised and responded to each of them one by one.
>
> `Q1: While this is not a big deal, the paper relies mostly on existing methodologies.`
>
> Thank you very much for recognizing our work! As you mentioned, we also believe that the appropriate organic combination of existing tools to solve current open problems is a behavior that the community should encourage. Although LatentDiffuser is based on existing technologies, neither the latent diffusion model nor the exact energy-guided sampling method (CEP) has been fully applied to the planning paradigm in offline RL. We think it is unreasonable to design new tools or methods directly without verifying the effectiveness of existing tools.
>
> Furthermore, we would like to add that by combining the latent diffusion model and CEP with the TAP method, we can not only implement the planning process in TAP more holistically and highly performing but also improve the CEP algorithm. Specifically, CEP requires additional TD-learning for a Q-function to sample high-quality trajectories. In LatentDiffuser, due to the definition of latent actions, we can directly use the reward-to-go estimates output by the latent diffusion model to replace the Q-function in CEP, thereby reducing the training cost. Due to the flexibility of the 'latent action' definition, our framework can also efficiently plan in the raw action and skill spaces (see Appendix F).
>
> In summary, LatentDiffuser is not a simple rearrangement of existing technologies. Each tool has its necessity and rationality for implementing efficient planning, and we have also adapted and improved these tools in a task-oriented manner.
>
> `Q2: The overall notation, in particular where x and y get used, together with $\tau$, and state tuples, can be a bit confusing.`
>
> We apologize for any confusion caused by the misuse of symbols. In the original version, we used boldface x to represent the noised trajectory and the latent action. In the revised version, we have amended the symbols in Section 3, using $\tau_{k}$ to represent the noised trajectory directly. We hope that these modifications will make the manuscript more readable.
>
> `Q3: Can the authors comment on how the sequence length is determined? Is the sequence terminated externally? Small stuff`
>
> Thank you for your question. We are unsure what "sequence length" means in this context. We will explain all parts of the paper that involve sequences.
>
> 1. If you are referring to the horizon or episode length, the length is determined by the simulator, and when it is terminated is also decided by the simulator. Termination may occur due to reaching a termination condition (task completion or failure) or exceeding the maximum length and being truncated.
> 2. If you are referring to the timesteps corresponding to the latent action or latent steps, the length is a fixed hyperparameter. In all experiments, the default setting is 3. We also conducted ablation studies on this hyperparameter in Appendix G.
> 3. If you are referring to the length of the sub-trajectory planned during the planning phase (corresponding to the length of the latent action sequence in LatentDiffuser's energy-guided sampling), the length is also a fixed hyperparameter. In all experiments, the default setting is 40. We also conducted ablation studies on this hyperparameter in Appendix G.
> 4. If you are referring to the diffusion steps, the length is also a fixed hyperparameter. In all experiments, the default setting is 100. We also conducted ablation studies on this hyperparameter in Appendix G.
>
> `Q4: A reference for the proof for theorem 1 is missing.`
>
> We apologize for our oversight. In the revised version, we have added references to the proof process.
>
> `Q5: Is the text inconsistent with Fig2? It seems like the initial state in Fig 2 is labeled s0, while in the text, there is only ever reference to s1.`
>
> We apologize. This was indeed an oversight on our part. In the revised version, we have updated Figure 2 in the main text and Figure 5 in the Appendix so that the indices of the sequences start from 1.
>
> `Q6: On page 5 typo "emploies".`
>
> Thank you for your thorough review! We have polished the entire text and corrected the typos in the revised version.

---

> ### Author Response · Authors · 2023-11-21
> **Have we addressed your concerns?**
>
> Thanks again for your time and effort in reviewing our paper! As the discussion period is coming to a close, we would like to know if we have resolved your concerns expressed in the original reviews. We remain open to any further feedback and are committed to making additional improvements if needed. If you find that these concerns have been resolved, we would be grateful if you would consider reflecting this in your rating of our paper : )

---

> > ### Comment · Reviewer_qvER · 2023-11-22
> >
> > Thank you for taking the time to respond, and making an effort in the rebuttal period. You have addressed the (minor) concerns I have had, and no fundamental issues seem to have come up through the review process. I have updated my score.

---

> > > ### Author Response · Authors · 2023-11-23
> > > **Respect and Best Wishes from the authors**
> > >
> > > We deeply appreciate the effort and time you've invested in the discussions, and we are delighted to receive your approval! Thank you for recognizing our efforts, and we also believe that the hard work we have done is worthwhile.
> > >
> > > The above inspiring discussion has greatly improved the quality of our paper. Thank you very much. We also enjoy the inspiring and insightful discussions with you!
> > >
> > > -- Respect and Best Wishes from the authors ^_^

---

### Official Review · Reviewer_kDwN · 2023-11-01

**Soundness:** 3 good
**Presentation:** 3 good
**Contribution:** 3 good
**Rating:** 6
**Confidence:** 2

**Summary:**

This paper presents a latent-action-based methods for reinforcement learning. This paper shows a unified framework for
continuous latent action space representation learning and planning by leveraging latent, score-based diffusion models, brining theoretical connections. The experiments from the offline RL settings are pretty solid.

**Strengths:**

1. The paper is well presented. The algorithm is straightforward and easy to understand.
2. The paper includes some connection to the theoretical analysis.

**Weaknesses:**

1. The paper could be stronger if the author could provide some visualization/analysis what the latent diffusers learned

**Questions:**

I think in general the paper is quite self-contained

---

> ### Author Response · Authors · 2023-11-15
>
> We greatly appreciate your taking the time to carefully review our paper and provide valuable suggestions and constructive feedback. This has significantly helped us improve the rigor and completeness of our paper. Additionally, we are grateful for your recognition of our work! Below, we have organized your questions and will address them individually.
>
> `Q1: The paper could be stronger if the author could provide some visualization/analysis what the latent diffusers learned.`
>
> Thank you very much for your suggestion. We have visualized the learned latent action space in Appendix H in the revised version. The specific analysis is as follows.
>
> To gain a more intuitive understanding of the latent action space learned by LatentDiffuser, this section presents a visualization of the latent actions and the corresponding trajectories obtained by decoding them. Specifically, we use the fully trained LatentDiffuser in the Hopper task to sample $5$ trajectories and apply the t-SNE method to reduce the dimensionality of the latent actions associated with these $5$ trajectories for visualization, as shown in Figure 8.
> In the visualization of the trajectories, a random trajectory is selected. To facilitate presentation (due to the similarity in the robot shape at adjacent timesteps), we downsampled this trajectory by taking one latent action for every $5$ latent action and overlaid the trajectory images obtained by decoding the adjacent $3$ latent actions with different opacities to achieve the trajectory displayed in Figure 8. It can be seen from the figure that the latent action space learned by LatentDiffuser is a more compact action space, which to some extent, has learned a specific type of macro-action or "skill."
>
> We hope these visualization results give you a more intuitive understanding of our algorithm. Once again, we appreciate your recognition of our work!

---

> ### Author Response · Authors · 2023-11-21
> **Have we addressed your concerns?**
>
> Thanks again for your time and effort in reviewing our paper! As the discussion period is coming to a close, we would like to know if we have resolved your concerns expressed in the original reviews. We remain open to any further feedback and are committed to making additional improvements if needed. If you find that these concerns have been resolved, we would be grateful if you would consider reflecting this in your rating of our paper :)

---

> ### Comment · Reviewer_kDwN · 2023-11-23
> **Thanks**
>
> I would like to thank the author for addressing my concerns. I will keep my score for acceptance.

---

> ### Author Response · Authors · 2023-11-23
> **Best wishes from all the authors**
>
> Thank you very much for recognizing our work! We appreciate the time and effort you've invested in reviewing our manuscript.
>
> --Best wishes from all the authors :)

---

### Author Response · Authors · 2023-11-15
**Common Response (Revision Uploaded)**

We would like to express our sincere gratitude to all the reviewers for taking the time out of their busy schedules to carefully review our paper and provide a series of constructive comments and suggestions. These comments and suggestions have been enormously helpful in enhancing the rigor and comprehensiveness of our paper. We have now submitted the revised version, in which all additions and modifications are highlighted in $\textcolor{red}{red}$ (except for some typo corrections that are not highlighted). Any numbered references in our individual responses to each reviewer pertain to the revised version, rather than the original version. We look forward to hearing back from the reviewers!

---

### Meta-Review · Area_Chair_yCzR · 2023-12-08

**Metareview:**

This paper presents a new approach to offline RL that involves using a latent representation of the path, and sampling latent variables using a conditional diffusion model.

Effectiveness demonstrated on standard RL continuous control tasks from D4RL, demonstrating that performance is on par or better than baselines for tasks, especially with high-dimensional continuous action spaces.

Reviewers initially raised some concerns about clarity of certain technical points in the formulation and evaluation and also offered suggestions to better differentiate the proposed approach from prior work. These were mostly addressed during the rebuttal period and the conclusion is to recommend accepting this paper.

**Justification For Why Not Higher Score:**

Several reviewers noted that similar ideas have already been explored in other settings, so the novelty is limited.

**Justification For Why Not Lower Score:**

The idea is sufficiently novel and the experimental results sufficiently strong that I expect the paper to be of interest to the communities working on offline and planning with conditional generative models.

---

### Decision · Program_Chairs · 2024-01-16

Accept (poster)